# The Cold Posterior Effect Indicates Underfitting, and Cold Posteriors Represent a Fully Bayesian Method to Mitigate It

**Yijie Zhang**  *yizh@di.ku.dk*
*University of Copenhagen*
*Novo Nordisk A/S*

**Yi-Shan Wu**  *yswu@imada.sdu.dk*
*University of Southern Denmark*

**Luis A. Ortega**  *luis.ortega@uam.es*
*Autonomous University of Madrid*

**Andrés R. Masegosa**  *arma@cs.aau.dk*
*Aalborg University*

**Reviewed on OpenReview:** *https://openreview.net/forum?id=GZORXGxHHT*

## Abstract

The cold posterior effect (CPE) (Wenzel et al., 2020) in Bayesian deep learning shows that, for posteriors with a temperature $T < 1$, the resulting posterior predictive could have better performance than the Bayesian posterior ($T = 1$). As the Bayesian posterior is known to be optimal under perfect model specification, many recent works have studied the presence of CPE as a model misspecification problem, arising from the prior and/or from the likelihood. In this work, we provide a more nuanced understanding of CPE as we show that *misspecification leads to CPE only when the resulting Bayesian posterior underfits*. In fact, we theoretically show that if there is no underfitting, there is no CPE. Furthermore, we show that these *tempered posteriors* with $T < 1$ are indeed proper Bayesian posteriors with a different combination of likelihoods and priors parameterized by $T$. This observation validates the adjustment of the temperature hyperparameter $T$ as a straightforward approach to mitigate underfitting in the Bayesian posterior. In essence, we show that by fine-tuning the temperature $T$ we implicitly utilize alternative Bayesian posteriors, albeit with less misspecified likelihood and prior distributions. The code for replicating the experiments can be found at https://github.com/pyijiezhang/cpe-underfit.

## 1 Introduction

In Bayesian deep learning, the cold posterior effect (CPE) (Wenzel et al., 2020) refers to the phenomenon in which if we artificially "temper" the posterior by either $p(\boldsymbol{\theta}|D) \propto (p(D|\boldsymbol{\theta})p(\boldsymbol{\theta}))^{1/T}$ or $p(\boldsymbol{\theta}|D) \propto p(D|\boldsymbol{\theta})^{1/T}p(\boldsymbol{\theta})$ with a temperature $T < 1$, the resulting posterior enjoys better predictive performance than the standard Bayesian posterior (with $T = 1$). The discovery of the CPE has sparked debates in the community about its potential contributing factors.

If the prior and likelihood are properly specified, the Bayesian solution (i.e., $T = 1$) should be optimal (Gelman et al., 2013), assuming approximate inference is properly working. Hence, the presence of the CPE implies either the prior (Wenzel et al., 2020; Fortuin et al., 2022), the likelihood (Aitchison, 2021; Kapoor et al., 2022), or both are misspecified. This has been, so far, the main argument of many works trying to explain the CPE.

One line of research examines the impact of the prior misspecification on the CPE (Wenzel et al., 2020; Fortuin et al., 2022). The priors of modern Bayesian neural networks are often selected for tractability. Consequently, the quality of the selected priors in relation to the CPE is a natural concern. Previous research has revealed that adjusting priors can help alleviate the CPE in certain cases (Adlam et al., 2020; Zeno et al., 2021), there are instances where the effect persists despite such adjustments (Fortuin et al., 2022). Some studies even show that the role of priors may not be critical (Izmailov et al., 2021). Therefore, the impact of priors on the CPE remains an open question.

Furthermore, the influence of likelihood misspecification on CPE has also been investigated (Aitchison, 2021; Noci et al., 2021; Kapoor et al., 2022; Fortuin et al., 2022), and has been identified to be particularly relevant in curated datasets (Aitchison, 2021; Kapoor et al., 2022). Several studies have proposed alternative likelihood functions to address this issue and successfully mitigate the CPE (Nabarro et al., 2022; Kapoor et al., 2022). However, the underlying relation between the likelihood and CPE remains a partially unresolved question. Notably, the CPE usually emerges when data augmentation (DA) techniques are used (Wenzel et al., 2020; Izmailov et al., 2021; Fortuin et al., 2022; Noci et al., 2021; Nabarro et al., 2022; Kapoor et al., 2022). A popular hypothesis is that using DA implies the introduction of a randomly perturbed log-likelihood, which lacks a clear interpretation as a valid likelihood function (Wenzel et al., 2020; Izmailov et al., 2021). However, Nabarro et al. (2022) demonstrates that the CPE persists even when a proper likelihood function incorporating DA is defined. Therefore, further investigation is needed to fully understand their relationship.

Other works argued that CPE could mainly be an artifact of inaccurate approximate inference methods, especially in the context of neural networks, where the posteriors are extremely high dimensional and complex (Izmailov et al., 2021). However, many of the previously mentioned works have also found setups where the CPE either disappears or is significantly alleviated through the adoption of better priors and/or better likelihoods with approximate inference methods. In these studies, the same approximate inference methods were used to illustrate, for example, how using a standard likelihood function leads to the observation of CPE and how using an alternative likelihood function removes it (Aitchison, 2021; Noci et al., 2021; Kapoor et al., 2022). In other instances, under the same approximate inference scheme, CPE is observed when using certain types of priors but it is strongly alleviated when an alternative class of priors is utilized (Wenzel et al., 2020; Fortuin et al., 2022). Therefore, there is compelling evidence suggesting that approximate methods are not, at least, a necessary condition for the CPE.

This study, both theoretically and empirically, demonstrates that the presence of the cold posterior effect (CPE) implies the existence of underfitting; in other words, *if there is no underfitting, there is no CPE.* Integrating this perspective with previous findings suggesting that CPE indicates misspecified likelihood, prior, or both (Gelman et al., 2013), we conclude that CPE implies both misspecification and underfitting. Consequently, mitigating CPE necessitates addressing both aspects. Notably, simplifying the issue by solely focusing on misspecification is insufficient, as misspecification can lead Bayesian methods to both underfitting and overfitting (Domingos, 2000; Immer et al., 2021; Kapoor et al., 2022); CPE only arises when underfitting occurs. This study thus offers a nuanced perspective on the factors contributing to CPE. Additionally, by building on Zeno et al. (2021), we show how tempered posteriors represent proper Bayesian posteriors under different likelihood and prior distributions, jointly parameterized by the temperature parameter $T$. Consequently, by adjusting $T$, we effectively identify Bayesian posteriors with less misspecified likelihood and prior distributions, leading to a more accurate representation of the training data and improved generalization performance. Furthermore, we delve into the relationship between prior/likelihood misspecification, data augmentation, approximate inference, and CPE, offering insights into potential strategies for addressing these issues.

**Contributions**  **(i)** We theoretically demonstrate that the presence of the CPE implies the Bayesian posterior is underfitting in Section 3. **(ii)** We show in a more general case that any tempered posterior is a proper Bayesian posterior with an alternative likelihood and prior distribution in Section 4, extending Zeno et al. (2021) in the case of classification. **(iii)** We show in Section 5 that likelihood misspecification and prior misspecification result in CPE only if they also induce underfitting. Furthermore, the tempered posteriors offer an effective and well-founded Bayesian mechanism to address the underfitting problem. **(iv)** Finally, we show that data augmentation results in stronger CPE because it induces a stronger underfitting of the

Bayesian posterior in Section 6. In conclusion, our theoretical analysis reveals that the occurrence of the CPE signifies underfitting of the Bayesian posterior. Also, fine-tuning the temperature in tempered posteriors offers a well-founded and effective Bayesian approach to mitigate the issue. Furthermore, our work aims to settle the debate surrounding CPE and its implications for Bayesian principles, specifically within the context of deep learning.

## 2 Background

### 2.1 Notation and assumptions

Let us start by introducing basic notation. Consider a supervised learning problem with the sample space $\mathcal{Y} \times \mathcal{X}$. In this work, we consider two cases: when $\mathcal{Y}$ is a finite set, corresponding to a supervised classification problem, and when $\mathcal{Y}$ is a subset of $\mathbb{R}$, corresponding to a regression problem. For simplicity, we also assume that $\mathcal{X}$ is a subset of $\mathbb{R}^d$. Let the training set $D = (\boldsymbol{Y}, \boldsymbol{X})$, where $\boldsymbol{Y}$ denotes the set of output entries and $\boldsymbol{X}$ denotes the set of input entries. If $D$ consists $n$ pairs of samples, we denote $D = \{(\boldsymbol{y}_i, \boldsymbol{x}_i)\}_{i=1}^n$.

We assume a family of probabilistic models parameterized by $\boldsymbol{\Theta}$, where each $\boldsymbol{\theta} \in \boldsymbol{\Theta}$ defines a conditional probability distribution for a sample $(\boldsymbol{y}, \boldsymbol{x})$, denoted by $p(\boldsymbol{y}|\boldsymbol{x}, \boldsymbol{\theta})$. The standard metric to measure the quality of a probabilistic model $\boldsymbol{\theta}$ on a sample $(\boldsymbol{y}, \boldsymbol{x})$ is the (negative) log-loss $-\ln p(\boldsymbol{y}|\boldsymbol{x}, \boldsymbol{\theta})$. The expected (or population) loss of a probabilistic model $\boldsymbol{\theta}$ is defined as $L(\boldsymbol{\theta}) = \mathbb{E}_{(\boldsymbol{y},\boldsymbol{x})\sim\nu}[-\ln p(\boldsymbol{y}|\boldsymbol{x}, \boldsymbol{\theta})]$, where $\nu$ denotes the unknown data-generating distribution $\nu$ on $\mathcal{Y} \times \mathcal{X}$. The empirical loss of the model $\boldsymbol{\theta}$ on the data $D$ is defined as $\hat{L}(D, \boldsymbol{\theta}) = -\frac{1}{n}\ln p(\boldsymbol{Y}|\boldsymbol{X}, \boldsymbol{\theta}) = -\frac{1}{n}\sum_{i\in[n]}\ln p(\boldsymbol{y}_i|\boldsymbol{x}_i, \boldsymbol{\theta})$. In this work, we assume that the likelihood function fully factorizes, i.e., $p(\boldsymbol{Y}|\boldsymbol{X}, \boldsymbol{\theta}) = \prod_{(\boldsymbol{y},\boldsymbol{x})\in D} p(\boldsymbol{y}|\boldsymbol{x}, \boldsymbol{\theta})$. We might use the notation $p(D|\boldsymbol{\theta})$ for $p(\boldsymbol{Y}|\boldsymbol{X}, \boldsymbol{\theta})$ in the presentation when the roles of input/output in the samples are not important in the context. Also, if it induces no ambiguity, we use $\mathbb{E}_\nu[\cdot]$ as a shorthand for $\mathbb{E}_{(\boldsymbol{y},\boldsymbol{x})\sim\nu}[\cdot]$.

### 2.2 (Generalized) Bayesian learning

In Bayesian learning, we learn a probability distribution $\rho(\boldsymbol{\theta}|D)$, often called a posterior, over the parameter space $\boldsymbol{\Theta}$ from the training data $D$. Given a new input $\boldsymbol{x}$, the posterior $\rho$ makes the prediction about $\boldsymbol{y}$ through (an approximation of) *Bayesian model averaging (BMA)* $p(\boldsymbol{y}|\boldsymbol{x}, \rho) = \mathbb{E}_{\boldsymbol{\theta}\sim\rho}[p(\boldsymbol{y}|\boldsymbol{x}, \boldsymbol{\theta})]$, where the posterior $\rho$ is used to combine the predictions of the models. Again, if it induces no ambiguity, we use $\mathbb{E}_\rho[\cdot]$ as a shorthand for $\mathbb{E}_{\boldsymbol{\theta}\sim\rho}[\cdot]$. The predictive performance of such BMA is usually measured by the Bayes loss, defined by

$$B(\rho) = \mathbb{E}_\nu[-\ln \mathbb{E}_\rho[p(\boldsymbol{y}|\boldsymbol{x}, \boldsymbol{\theta})]]. \tag{1}$$

For some $\lambda > 0$ and a prior $p(\boldsymbol{\theta})$, the so-called *tempered posteriors* (or the generalized Bayes posterior) (Barron & Cover, 1991; Zhang, 2006; Bissiri et al., 2016; Grünwald & van Ommen, 2017), are defined as a probability distribution

$$p_\lambda(\boldsymbol{\theta}|D) \propto p(\boldsymbol{Y}|\boldsymbol{X}, \boldsymbol{\theta})^\lambda p(\boldsymbol{\theta}). \tag{2}$$

Note that when $\lambda \neq 1$, $\int p(\boldsymbol{Y}|\boldsymbol{X}, \boldsymbol{\theta})^\lambda d\boldsymbol{Y}$ might not be 1 in general. An implicit assumption is that $p_\lambda(\boldsymbol{\theta}|D)$ is a *proper distribution*, meaning the normalization constant is finite. In supervised classification problems, this is always the case because $p(\boldsymbol{Y}|\boldsymbol{X}, \boldsymbol{\theta}) \leq 1$. Consequently, for any $\lambda > 0$, we have $1 = \int p(\boldsymbol{\theta})\,d\boldsymbol{\theta} > \int p(\boldsymbol{Y}|\boldsymbol{X}, \boldsymbol{\theta})^\lambda p(\boldsymbol{\theta})\,d\boldsymbol{\theta}$. Thus, the tempered posteriors are always a proper distribution in supervised classification problems.

Even though many works on CPE use the parameter $T = 1/\lambda$ instead, we adopt $\lambda$ in the rest of the work for the convenience of derivations. Therefore, the CPE ($T < 1$) corresponds to when $\lambda > 1$. We also note that while some works study CPE with a full-tempering posterior, where the prior is also tempered, many works also find CPE for likelihood-tempering posterior (see (Wenzel et al., 2020) and the references therein). Also, with some widely chosen priors (e.g., zero-centered Gaussian priors), the likelihood-tempering posteriors are equivalent to full-tempering posteriors with rescaled prior variances (Aitchison, 2021; Bachmann et al., 2022).

When $\lambda = 1$, the tempered posterior equals the (standard) Bayesian posterior. The tempered posterior can be obtained by optimizing a generalization of the so-called (generalized) ELBO objective (Alquier et al., 2016;

Higgins et al., 2017), which, for convenience, we write as follows:

$$p_\lambda(\boldsymbol{\theta}|D) = \arg\min_\rho \mathbb{E}_\rho[-\ln p(D|\boldsymbol{\theta})] + \frac{1}{\lambda}\,\mathrm{KL}(\rho(\boldsymbol{\theta}|D), p(\boldsymbol{\theta}))\,. \tag{3}$$

The first term is known as the (un-normalized) *reconstruction error* or the empirical Gibbs loss of the posterior $\rho$ on the data $D$, denoted as $\hat{G}(\rho, D) = \mathbb{E}_\rho[-\frac{1}{n}\ln p(D|\boldsymbol{\theta})]$, which further equals to $\mathbb{E}_\rho[\hat{L}(D, \boldsymbol{\theta})]$. Therefore, it is often used as the *training loss* in Bayesian learning (Morningstar et al., 2022). The second term is a Kullback-Leibler divergence between the posterior $\rho(\boldsymbol{\theta}|D)$ and the prior $p(\boldsymbol{\theta})$ scaled by a hyper-parameter $\lambda$.

As it induces no ambiguity, we will use $p_\lambda$ as a shorthand for $p_\lambda(\boldsymbol{\theta}|D)$. So, for example, $B(p_\lambda)$ would refer to the expected Bayes loss of the tempered-posterior $p_\lambda(\boldsymbol{\theta}|D)$. In the rest of this work, we will interpret the CPE as how changes in the parameter $\lambda$ affect the *test error* and the *training error* of $p_\lambda$ or, equivalently, the Bayes loss $B(p_\lambda)$ and the empirical Gibbs loss $\hat{G}(p_\lambda, D)$.

## 3 The presence of the CPE implies underfitting

> **Section Overview**
>
> We present a definition of the Cold Posterior Effect (CPE) (Definition 1) and show that the presence of CPE indicates the Bayesian posterior is underfitting, where both the testing loss (Definition 1) and training loss (Proposition 1) can be improved at the same time by decreasing the temperature $T$ (increasing $\lambda$). We also present the necessary condition of the CPE in Proposition 3 and the case when Bayesian posterior is optimal in Theorem 4.

A standard understanding for underfitting refers to a situation when the trained model cannot properly capture the relationship between input and output in the data-generating process, resulting in high errors on both the training data and testing data. In the context of highly flexible model classes such as neural networks, underfitting refers to a scenario where the trained model exhibits (much) higher training and testing losses compared to what is achievable. Essentially, it means that there exists another model in the model class that achieves lower training and testing losses simultaneously. In the context of Bayesian inference, we argue that the Bayesian posterior is underfitting if there exists another posterior distribution with lower empirical Gibbs and Bayes losses at the same time. In fact, we will show later in Section 4 that such a posterior is essentially another *Bayesian posterior but with a different prior and likelihood function*. Before delving into that, we focus on characterizing the cold posterior effect (CPE) and its connection to underfitting.

As previously discussed, the CPE describes the phenomenon of getting better predictive performance when we make the parameter of the tempered posterior, $\lambda$, higher than 1. The next definition introduces a formal characterization. *We do not claim this is the best possible formal characterization.* However, through the rest of the paper, we will show that this simple characterization is enough to understand the relationship between CPE and underfitting.

**Definition 1.** *We say there is a CPE for Bayes loss if and only if the derivative of the Bayes loss of the posterior $p_\lambda$, $B(p_\lambda)$, evaluated at $\lambda = 1$ is negative. That is,*

$$\frac{d}{d\lambda}B(p_\lambda)_{|\lambda=1} < 0\,, \tag{4}$$

*where the magnitude of the derivative $\frac{d}{d\lambda}B(p_\lambda)_{|\lambda=1}$ defines the strength of the CPE.*

According to the above definition, a (relatively large) negative derivative $\frac{d}{d\lambda}B(p_\lambda)_{|\lambda=1}$ implies that by making $\lambda$ slightly greater than 1, we will have a (relatively large) reduction in the Bayes loss with respect to the Bayesian posterior. Note that if the derivative $\frac{d}{d\lambda}B(p_\lambda)_{|\lambda=1}$ is not relatively large and negative, then we can not expect a relatively large reduction in the Bayes loss and, in consequence, the CPE will not be significant. Obviously, this formal definition could also be extended to other specific $\lambda$ values different from 1, or even consider some aggregation over different $\lambda > 1$ values. We will stick to this definition because it is simpler, and

the insights and conclusions extracted here can be easily extrapolated to other similar definitions involving the derivative of the Bayes loss.

Next, we present another critical observation. We postpone the proofs in this section to Appendix A.

**Proposition 2.** *The derivative of the empirical Gibbs loss of the tempered posterior $p_\lambda$ satisfies*

$$\forall \lambda \geq 0 \quad \frac{d}{d\lambda}\hat{G}(p_\lambda, D) = -\mathbb{V}_{p_\lambda}\big(\ln p(D|\boldsymbol{\theta})\big) \leq 0 , \tag{5}$$

*where $\mathbb{V}(\cdot)$ denotes the variance.*

As shown in Proposition 7 in Appendix A, to achieve $\mathbb{V}_{p_\lambda}\big(\ln p(D|\boldsymbol{\theta})\big) = 0$, we need $p_\lambda(\boldsymbol{\theta}|D) = p(\boldsymbol{\theta})$, implying that the data has no influence on the posterior. In consequence, in practical scenarios, $\mathbb{V}_{p_\lambda}\big(\ln p(D|\boldsymbol{\theta})\big)$ will always be greater than zero. Thus, increasing $\lambda$ will monotonically reduce the empirical Gibbs loss $\hat{G}(p_\lambda, D)$ (i.e., the *train error*) of $p_\lambda$. The next result also shows that the empirical Gibbs loss of the Bayesian posterior $\hat{G}(p_{\lambda=1})$ cannot reach its minimum to observe the CPE.

**Proposition 3.** *A necessary condition for the presence of the CPE, as defined in Definition 1, is that*

$$\hat{G}(p_{\lambda=1}, D) > \min_{\boldsymbol{\theta}} -\ln p(D|\boldsymbol{\theta}) .$$

**Insight 1.** *Definition 1 in combination with Proposition 2 shows if the CPE is present, by making $\lambda > 1$, the test loss $B(p_\lambda)$ and the empirical Gibbs loss $\hat{G}(p_\lambda, D)$ will be reduced at the same time. Furthermore, Proposition 3 states that the Bayesian posterior still has room to fit the training data further (e.g., by placing more probability mass on the maximum likelihood estimator). We hence deduce that the presence of CPE implies that the original Bayesian posterior ($\lambda = 1$) underfits. This conclusion arises because there exists another Bayesian posterior (i.e, $p_\lambda(\boldsymbol{\theta}|D)$ with $\lambda > 1$) that has lower training (Proposition 3) and testing (Definition 1) loss at the same time. Further elaboration on the nature of $p_\lambda(\boldsymbol{\theta}|D)$ as another Bayesian posterior will be provided later in Section 4. In short, if there is CPE, the original Bayesian posterior is underfitting. Or, equivalently, if the original Bayesian posterior does not underfit, there is no CPE.*

However, a final question arises: when is $\lambda = 1$ (the original Bayesian posterior of interest) *optimal*? More precisely, when does the derivative of the Bayes loss with respect to $\lambda$ evaluated at $\lambda = 1$ become zero ($\frac{d}{d\lambda}B(p_\lambda)_{|\lambda=1} = 0$)? This would imply that neither (infinitesimally) increasing nor decreasing $\lambda$ changes the predictive performance. We will see that this condition is closely related to the situation that updating such a Bayesian posterior with more data does not enhance its fit to the original training data better. In other words, the extra information about the data-generation process does not provide the Bayesian posterior with better performance on the originally provided training data.

We start by denoting $\tilde{p}_\lambda(\boldsymbol{\theta}|D, (\boldsymbol{y}, \boldsymbol{x}))$ as the distribution obtained by updating the posterior $p_\lambda(\boldsymbol{\theta}|D)$ with one new sample $(\boldsymbol{y}, \boldsymbol{x})$, i.e., $\tilde{p}_\lambda(\boldsymbol{\theta}|D, (\boldsymbol{y}, \boldsymbol{x})) \propto p(\boldsymbol{y}|\boldsymbol{x}, \boldsymbol{\theta})p_\lambda(\boldsymbol{\theta}|D)$. And we also denote $\bar{p}_\lambda$ as the distribution resulting from averaging $\tilde{p}_\lambda(\boldsymbol{\theta}|D, (\boldsymbol{y}, \boldsymbol{x}))$ over different *unseen* samples from the data-generating distribution $(\boldsymbol{y}, \boldsymbol{x}) \sim \nu(\boldsymbol{y}, \boldsymbol{x})$:

$$\bar{p}_\lambda(\boldsymbol{\theta}|D) = \mathbb{E}_\nu\left[\tilde{p}_\lambda(\boldsymbol{\theta}|D, (\boldsymbol{y}, \boldsymbol{x}))\right]. \tag{6}$$

In this sense, $\bar{p}_\lambda$ represents how the posterior $p_\lambda$ would be, on average, after being updated with a new sample from the data-generating distribution. This updated posterior contains a bit more information about the data-generating distribution, compared to $p_\lambda$. Using the updated posterior $\bar{p}_\lambda$, the following result introduces a characterization of the *optimality* of the original Bayesian posterior.

**Theorem 4.** *The derivative of the Bayes loss at $\lambda = 1$ is null, i.e., $\frac{d}{d\lambda}B(p_\lambda)_{|\lambda=1} = 0$, if and only if,*

$$\hat{G}(p_{\lambda=1}, D) = \hat{G}(\bar{p}_{\lambda=1}, D) .$$

**Insight 2.** *The original Bayesian posterior of interest is optimal if after updating it using the procedure described in Equation 6, or in other words, after exposing the Bayesian posterior to more data from the data-generating distribution, the empirical Gibbs loss over the initial training data remains unchanged.*

We will give examples that empirically illustrate Theorem 4 and the induced insight later in Section 5.5.

## 4 Tempered posteriors are Bayesian posteriors

> **Section Overview**
>
> By extending Zeno et al. (2021) on classification only, we show in general that tempered posteriors are proper Bayesian posteriors with an alternative combination of likelihood and prior functions parameterized by $\lambda$. Thus, the occurrence of CPE can be explained within the Bayesian framework.
>
> - We provide two examples to show how $\lambda$ influences the new likelihoods in Section 4.1 and two examples to show how $\lambda$ influences the new priors in Section 4.2.
>
> - We show in Section 4.3 that the generalized ELBOs are also proper ELBOs.
>
> - We expand the discussion of the implications in Section 4.4.

As previously discussed, the CPE phenomenon involves achieving improved predictive accuracy by employing a tempered posterior. A potential criticism is that this tempered posterior does not strictly adhere to the principles of a proper Bayesian posterior because the tempered likelihood, $P(D|\boldsymbol{\theta})^\lambda$ fails to meet the criteria of a proper likelihood function when $\lambda \neq 1$ (i.e., $\int P(D|\boldsymbol{\theta})^\lambda dD \neq 1$ when $\lambda \neq 1$). However, as previously discussed by Zeno et al. (2021), this tempered posterior effectively serves as a *proper Bayesian posterior* with a combination of *new likelihood and prior functions*. We extend this result beyond classification to our Proposition 5, proved in Appendix B.1.

Before delving into the description of the new likelihood and prior functions, it is essential to acknowledge a fundamental aspect. Given a labeled dataset $D = (\boldsymbol{X}, \boldsymbol{Y})$ and the conditional likelihood associated to a classification model, the application of Bayes' theorem naturally results in the following Bayesian posterior:

$$p(\boldsymbol{\theta}|\boldsymbol{X}, \boldsymbol{Y}) \propto p(\boldsymbol{Y}|\boldsymbol{X}, \boldsymbol{\theta})p(\boldsymbol{\theta}|\boldsymbol{X}),$$

where the prior over $\boldsymbol{\theta}$ is a *conditional prior* (Marek et al., 2024; Zeno et al., 2021) that depends on the unlabelled training data $\boldsymbol{X}$. However, specifying $p(\boldsymbol{\theta}|\boldsymbol{X})$ for a complex model, like a deep neural network, poses a significant challenge. Therefore, for practical purposes, nearly all existing works (Wenzel et al., 2020; Fortuin et al., 2022) assume $\boldsymbol{\theta}$ to be independent of $\boldsymbol{X}$, resulting in the simplified expression $p(\boldsymbol{\theta}|\boldsymbol{X}, \boldsymbol{Y}) \propto p(\boldsymbol{Y}|\boldsymbol{X}, \boldsymbol{\theta})p(\boldsymbol{\theta})$, where the prior over $\boldsymbol{\theta}$ is now an *unconditional prior*.

**Proposition 5.** *For any given dataset $D = (\boldsymbol{X}, \boldsymbol{Y})$ such that the likelihood fully factories $p(\boldsymbol{Y}|\boldsymbol{X}, \boldsymbol{\theta}) = \prod_{(\boldsymbol{y}, \boldsymbol{x}) \in D} p(\boldsymbol{y}|\boldsymbol{x}, \boldsymbol{\theta})$, and $\lambda > 0$, the tempered posterior defined in Equation 2 can be expressed as a Bayesian posterior with a new prior and likelihood function as follows:*

$$p_\lambda(\boldsymbol{\theta}|\boldsymbol{X}, \boldsymbol{Y}) \propto q(\boldsymbol{\theta}|\boldsymbol{X}, \lambda) \prod_{(\boldsymbol{y}, \boldsymbol{x}) \in D} q(\boldsymbol{y}|\boldsymbol{x}, \boldsymbol{\theta}, \lambda), \tag{7}$$

*where the new prior distribution $q(\boldsymbol{\theta}|\boldsymbol{X}, \lambda)$ and likelihood function $q(\boldsymbol{y}|\boldsymbol{x}, \boldsymbol{\theta}, \lambda)$ are defined as:*

$$q(\boldsymbol{\theta}|\boldsymbol{X}, \lambda) \propto p(\boldsymbol{\theta}) \prod_{\boldsymbol{x} \in \boldsymbol{X}} \int p(\boldsymbol{y}|\boldsymbol{x}, \boldsymbol{\theta})^\lambda d\boldsymbol{y}, \qquad q(\boldsymbol{y}|\boldsymbol{x}, \boldsymbol{\theta}, \lambda) = \frac{p(\boldsymbol{y}|\boldsymbol{x}, \boldsymbol{\theta})^\lambda}{\int p(\boldsymbol{y}|\boldsymbol{x}, \boldsymbol{\theta})^\lambda d\boldsymbol{y}}. \tag{8}$$

Note that the new conditional likelihood $q(\boldsymbol{y}|\boldsymbol{x}, \boldsymbol{\theta}, \lambda)$ and the new prior $q(\boldsymbol{\theta}|\boldsymbol{X}, \lambda)$ are both parametrized by the same $\lambda > 0$, and note that the prior only depends on the unlabelled training data $\boldsymbol{X}$.

Adlam et al. (2020) shows that in the specific scenario of Gaussian process regression (essentially our Bayesian linear regression example in Figure 2), any positive temperature aligns with a legitimate posterior under an adjusted unconditional prior. This can be seen as a special case of our argument.

In the rest of the section, we will discuss in Section 4.1 how the new likelihoods change with respect to $\lambda$, and in Section 4.2 how the new priors change with respect to $\lambda$. Additionally, besides demonstrating that tempered posteriors are proper Bayesian posteriors, we also show in Section 4.3 that the generalized ELBOs are also proper ELBOs. Lastly, we discuss the implications of the results in Section 4.4.

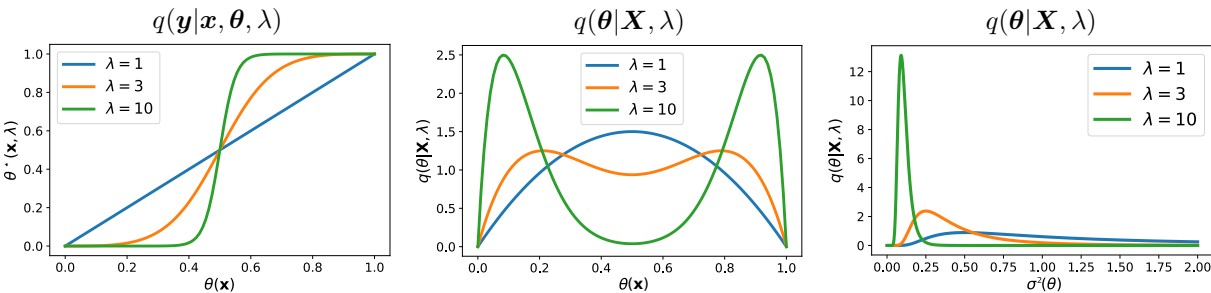

Figure 1: Illustration of the new likelihood $q(\boldsymbol{y}|\boldsymbol{x}, \boldsymbol{\theta}, \lambda)$ and priors $q(\boldsymbol{\theta}|\boldsymbol{X}, \lambda)$. In the left and middle figures, the original likelihood is in the form of the Bernoulli distribution. The left figure demonstrates the transformation from $\theta(\mathbf{x})$ to $\theta^\star(\boldsymbol{x}, \lambda) := \frac{\theta(\boldsymbol{x})^\lambda}{\theta(\boldsymbol{x})^\lambda + (1-\theta(\boldsymbol{x}))^\lambda}$. In the middle figure, we display a Beta-Binomial example, where the prior, initialized as a Beta distribution, is updated with a single Bernoulli-distributed sample. In the right figure, we display the new prior, initialized as an inverse-gamma prior and updated with a Gaussian likelihood with a single observation.

### 4.1 How $\lambda$ influences the new likelihoods

The next result, proved in Appendix B.2, shows that in supervised classification settings, higher $\lambda$ values induce new likelihood distributions with lower aleatoric uncertainty or, equivalently, lower Shannon entropy, denoted as $H(q(\boldsymbol{y}|\boldsymbol{x}, \boldsymbol{\theta}, \lambda)) := -\sum_{\boldsymbol{y} \in \mathcal{Y}} q(\boldsymbol{y}|\boldsymbol{x}, \boldsymbol{\theta}, \lambda) \ln q(\boldsymbol{y}|\boldsymbol{x}, \boldsymbol{\theta}, \lambda)$.

**Proposition 6.** *For any $\boldsymbol{\theta} \in \boldsymbol{\Theta}$, any $\boldsymbol{x} \in \mathcal{X}$, and any finite output set $\mathcal{Y}$, the entropy of the conditional likelihood $q(\boldsymbol{y}|\boldsymbol{x}, \boldsymbol{\theta}, \lambda)$ monotonically decreases with $\lambda > 0$, i.e.,*

$$\frac{d}{d\lambda} H(q(\boldsymbol{y}|\boldsymbol{x}, \boldsymbol{\theta}, \lambda)) \leq 0 \quad \forall \lambda > 0. \tag{9}$$

This result also holds for regression settings, where $\mathcal{Y} \subset \mathbb{R}^d$, under the differential entropy, assuming the Leibniz rule holds. See the proof for a detailed discussion on the matter.

We give two concrete examples, one in regression and one in classification, to illustrate the proposition.

**Regression Example** *Consider the case where the original likelihood is Gaussian, defined as $p(\boldsymbol{y}|\boldsymbol{x}, \boldsymbol{\theta}) = \mathcal{N}(\mu(\boldsymbol{x}, \boldsymbol{\theta}), \sigma^2(\boldsymbol{\theta}))$, where the variance is input-independent, as typically seen in many regression problems. Then, following Equation 8, the new likelihood corresponds to a scaling in the variance, given by $q(\boldsymbol{y}|\boldsymbol{x}, \boldsymbol{\theta}, \lambda) = \mathcal{N}(\mu(\boldsymbol{x}, \boldsymbol{\theta}), \frac{\sigma^2(\boldsymbol{\theta})}{\lambda^2})$. Thus, as $\lambda$ increases, the tempered likelihood $q(\boldsymbol{y}|\boldsymbol{x}, \boldsymbol{\theta}, \lambda)$ induces a proper Gaussian likelihood with reduced variance, i.e., a new likelihood with lower aleatoric uncertainty.*

**Classification Example** *Consider the case of a binary classification problem where the original conditional likelihood is Bernoulli, defined as $p(\boldsymbol{y}|\boldsymbol{x}, \boldsymbol{\theta}) = \theta(\boldsymbol{x})^y (1 - \theta(\boldsymbol{x}))^{1-y}$ with $y \in \{0, 1\}$ and the input-dependent parameter function $\theta(\boldsymbol{x}) \in [0, 1]$, which is usually implemented by a neural network with a softmax activation function in the last layer. Then, following Equation 8, the new conditional likelihood $q(\boldsymbol{y}|\boldsymbol{x}, \boldsymbol{\theta}, \lambda) = \theta^*(\boldsymbol{x}, \lambda)^y (1 - \theta^*(\boldsymbol{x}, \lambda))^{1-y}$ also follows a Bernoulli distribution with a different parameter function $\theta^*(\boldsymbol{x}, \lambda) = \frac{\theta(\boldsymbol{x})^\lambda}{\theta(\boldsymbol{x})^\lambda + (1-\theta(\boldsymbol{x}))^\lambda} \in [0, 1]$. The function $\theta^*(\boldsymbol{x}, \lambda)$ is displayed in Figure 1 (left). When $\lambda$ increases, the parameter function that defines the new Bernoulli likelihood becomes more extreme, resulting in a new likelihood with lower aleatoric uncertainty.*

In both cases, we see that as suggested by Proposition 6, as $\lambda$ increases, the new conditional likelihoods $q(\boldsymbol{y}|\boldsymbol{x}, \boldsymbol{\theta}, \lambda)$ have lower entropy, i.e., lower aleatoric uncertainty. In Sections 5.2 and 5.3, we will further explore the implications of this finding and its connection to existing literature.

### 4.2 How $\lambda$ influences the new priors

On the other hand, according to Proposition 5, using the tempered posteriors implies implicitly using the prior $q(\boldsymbol{\theta}|\boldsymbol{X}, \lambda)$. Such prior depends on the *unlabelled training data* $\boldsymbol{X}$. On top of that, the functional form

of the likelihood function is defined by the probabilistic model family through the term $\int p(\boldsymbol{y}|\boldsymbol{x}, \boldsymbol{\theta})^\lambda d\boldsymbol{y}$ for $\boldsymbol{x} \in \boldsymbol{X}$. Hence, models $\boldsymbol{\theta}$ that yield a large value for this term across most of the training data $\boldsymbol{x} \in \boldsymbol{X}$ will be assigned larger probability mass by the new prior. We will showcase this effect in both regression and binary classification problems. Moreover, we will see how the new prior $q(\boldsymbol{\theta}|\boldsymbol{X}, \lambda)$ with $\lambda > 1$ *favors those models within the model class that yield likelihoods with lower aleatoric uncertainty on the training data* $\boldsymbol{X}$.

**Regression Example**   *Consider the case where the original likelihood is Gaussian, defined as $p(\boldsymbol{y}|\boldsymbol{x}, \boldsymbol{\theta}) = \mathcal{N}(\mu(\boldsymbol{x}, \boldsymbol{\theta}), \sigma^2(\boldsymbol{\theta}))$, where the variance is input-independent, as typically seen in many regression problems. A common parametrization involves $\boldsymbol{\theta} = (\boldsymbol{w}, \gamma)$, where $\boldsymbol{w}$ refer to the weights of the neural network defining the function $\mu(\boldsymbol{x}, \boldsymbol{\theta})$ and $\gamma > 0$ is a parameter encoding the variance of the Gaussian likelihood such that $\sigma^2(\boldsymbol{\theta}) = \gamma$. The prior $p(\boldsymbol{\theta})$ is then defined as $p(\boldsymbol{\theta}) = p(\boldsymbol{w})p(\gamma)$, where $p(\boldsymbol{w})$ is usually a Gaussian distribution with a diagonal covariance matrix, and $p(\gamma)$ is usually defined in terms of an inverse-gamma distribution. Following Equation 8, the new prior would be expressed as $q(\boldsymbol{\theta}|\boldsymbol{X}, \lambda) = q(\boldsymbol{w}|\boldsymbol{X}, \lambda)q(\gamma|\boldsymbol{X}, \lambda)$, where each term*

$$q(\boldsymbol{w}|\boldsymbol{X}, \lambda) = p(\boldsymbol{w}) \quad , \quad q(\gamma|\boldsymbol{X}, \lambda) \propto p(\gamma)/\gamma^{n(\lambda-1)} .$$

*Figure 1 (right) plots the density of $q(\gamma|\boldsymbol{X}, \lambda)$ when only one data is observed, with various $\lambda > 1$ values when $p(\gamma)$ is an inverse-gamma prior. For larger $\lambda$ values, this new prior will assign more probability mass to models defining a likelihood with smaller variance or, equivalently, smaller aleatoric uncertainty.*

**Classification Example**   *Consider another case where the original conditional likelihood is Bernoulli, defined as $p(\boldsymbol{y}|\boldsymbol{x}, \boldsymbol{\theta}) = \theta(\boldsymbol{x})^y (1 - \theta(\boldsymbol{x}))^{1-y}$ with $y \in \{0, 1\}$ and $\theta(\boldsymbol{x}) \in [0, 1]$, as commonly used in binary classification problems. Also, take any prior $p(\boldsymbol{\theta})$. Then, following Equation 8, the new prior is expressed as*

$$q(\boldsymbol{\theta}|\boldsymbol{X}, \lambda) \propto p(\boldsymbol{\theta}) \prod_{\boldsymbol{x} \in \boldsymbol{X}} \left( \theta(\boldsymbol{x})^\lambda + (1 - \theta(\boldsymbol{x}))^\lambda \right) .$$

*Figure 1 (middle) illustrates the transformation of the prior for a Beta-Binomial model with a single training sample. Initially, the prior $p(\boldsymbol{\theta})$ is taken as a Beta distribution, while the likelihood of this single data is Bernoulli. As $\lambda \geq 1$ increases, the new prior assigns more probability mass to models where $\theta(\boldsymbol{x})$ is close to either 1 or 0. In other words, this new prior assigns more probability mass to models that assign more extreme probabilities to the training data (i.e., models with lower aleatoric uncertainty). Note that the prior does not consider how accurately these models classify the training data, but only the extremity of the probabilities assigned to the training data.*

We will discuss in Section 5.4 further implications of this finding and how it relates to the literature.

## 4.3   Generalized ELBOs are also proper ELBOs

Generalized ELBOs, characterized by scaling the KL divergence term using a hyper-parameter $\lambda$, have found widespread application in many studies (Wenzel et al., 2020). This popularity stems from the demonstrated ability to adjust $\lambda$ to improve the predictive accuracy of variational approximations:

$$q_\lambda^\star := \underset{r \in \Pi}{\arg\min} \, \mathbb{E}_r[-\ln p(D|\boldsymbol{\theta})] + \frac{1}{\lambda} \, \mathrm{KL}(r(\boldsymbol{\theta}), p(\boldsymbol{\theta})) , \tag{10}$$

where $\Pi$ defines the variational family. Critics have pointed out a flaw in the above generalized ELBO when $\lambda$ deviates from 1, as it no longer functions as a true lower bound for the marginal likelihood. However, Proposition 5 can be used to justify that such a variational posterior $q_\lambda^\star$ still emerges from minimizing a valid ELBO. Specifically, it is constructed based on the revised likelihood and prior functions as follows:

$$q_\lambda^\star = \underset{r \in \Pi}{\arg\min} \, \mathbb{E}_r[-\ln q(\boldsymbol{Y}|\boldsymbol{X}, \boldsymbol{\theta}, \lambda)] + \mathrm{KL}(r(\boldsymbol{\theta}), q(\boldsymbol{\theta}|\boldsymbol{X}, \lambda)) . \tag{11}$$

Consequently, this analysis shows that using generalized ELBOs as Equation 10 perfectly adheres to variational and Bayesian principles.

### 4.4 Insights and implications from the section

In this section, we show that employing tempered posteriors seamlessly fits within a Bayesian framework, which streamlines and enriches the use of diverse likelihood and prior functions.

With the characterization in Section 3, observing CPE implies that the tempered posterior, which implicitly employs the new likelihood and priors defined in Equation 8 is better specified in comparison to the original Bayesian posterior. The alignment with the underlying data-generating distribution is easily achieved by tempering. Consequently, tempered posteriors offer a simple, computationally efficient, and theoretically sound approach to mitigate the underfitting problem often encountered in contemporary Bayesian deep learning methods.

Furthermore, as discussed in Section 4.1 and Section 4.2, increasing $\lambda$ results in likelihoods with lower aleatoric uncertainty and priors that favor models yielding such likelihoods on the training data $X$. Therefore, the occurrence of CPE in contemporary Bayesian deep learning indicates that the models currently employed in the field often underfit the data by assuming models with too high aleatoric uncertainty. This strengthens our understanding of the CPE as a consequence of underfitting, resulting from poorly specified likelihood and prior functions.

We will further expand on and discuss how these implications relate to the literature in Section 5.

## 5 Likelihood misspecification, prior misspecification and the CPE

> **Section Overview**
>
> We relate our analysis in previous sections to the main arguments of CPE from the literature.
>
> - Section 5.1: we demonstrate with Bayesian linear regression that exact inference can also bring CPE, showing CPE is not merely a side effect of approximate inference in NNs.
>
> - Section 5.2: using the same regression examples, we show that model misspecification can lead to underfitting or overfitting. CPE arises specifically when misspecified likelihoods or priors cause underfitting, not just from misspecification alone.
>
> - Section 5.3: likelihood misspecification is often identified as a source of CPE in practice. We show it is because the standard softmax likelihood (high aleatoric uncertainty) is misspecified and underfits the data-generating process (curated data with low aleatoric uncertainty).
>
> - Section 5.4: we show prior misspecification leads to CPE if it induces underfitting. Using tempered posteriors implicitly defines better-specified conditional priors that alleviate it.
>
> - Section 5.5: we show that larger models have more flexibility to fit data, thereby mitigating underfitting and CPE. Conversely, with small models and abundant data, the Bayesian posterior may already fit the data optimally, thereby exhibiting minimal underfitting (by our definition) and CPE.

### 5.1 CPE, approximate inference, and NNs

As mentioned in the introduction, several works have discussed that CPE is an artifact of inappropriate approximate inference methods, especially in the context of the highly complex posterior that emerge from neural networks (Wenzel et al., 2020). There are occasions suggesting that if the approximate inference method is accurate enough, the CPE disappears (Izmailov et al., 2021). However, Proposition 2 shows that when $\lambda$ is made larger than 1, the *training loss* of the exact Bayesian posterior decreases; if the *test loss* decreases too, the exact Bayesian posterior underfits. It means that even if the inference method is accurate, we can still observe the CPE due to underfitting. In fact, Figure 2 shows examples of a Bayesian linear regression model learned on synthetic data. Here, the exact Bayesian posterior can be computed,

and it is clear from Figures 2c and 2d that the CPE can occur in Bayesian linear regression with exact inference. Although simple, the setting is articulated specifically to mimic the classification tasks using BNNs where CPE was observed. In particular, the linear model has more parameters than observations (i.e., it's overparameterized). We also note that Adlam et al. (2020) presents similar findings and observations for Gaussian process regression.

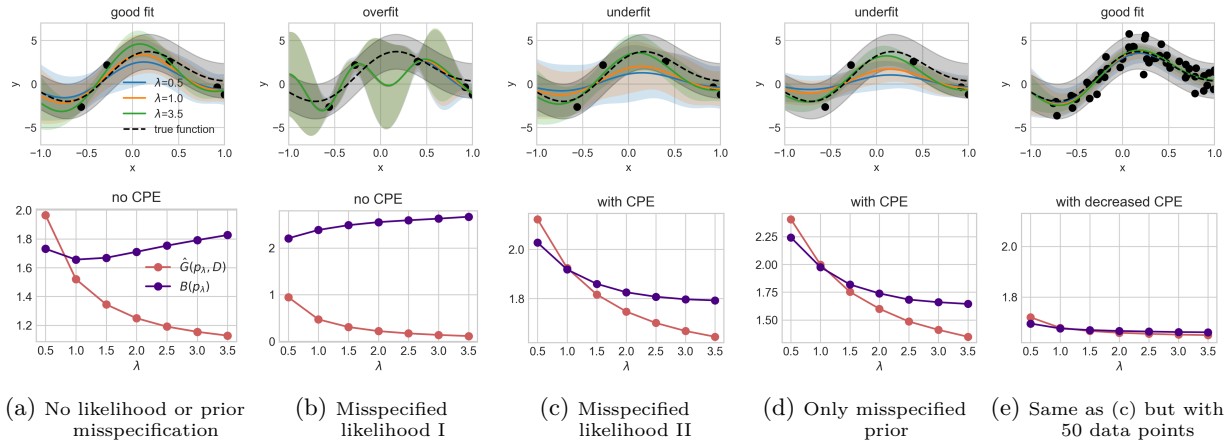

Figure 2: **1. The CPE occurs in Bayesian linear regression with exact inference. 2. Model misspecification can lead to overfitting and to a "warm" posterior effect (WPE).** Every column displays a specific setting, as indicated in the caption. The first row shows exact Bayesian posterior predictive fits for three different values of the tempering parameter $\lambda$. The second row shows the Gibbs loss $\hat{G}(p_\lambda, D)$ (aka training loss) and the Bayes loss $B(p_\lambda)$ (aka testing loss) with respect to $\lambda$. The experimental details are given in Appendix D.

## 5.2 Model misspecification, CPE, and underfitting

Prior and/or likelihood misspecification can lead Bayesian methods to both underfitting and overfitting, as widely discussed in the literature (Domingos, 2000; Immer et al., 2021; Kapoor et al., 2022). We illustrate this using a Bayesian linear regression model: Figures 2c and 2d show how the Bayesian posterior underfits due to likelihood and prior misspecification, respectively. On the other hand, Figure 2b showcases a scenario where likelihood misspecification can perfectly lead to overfitting as well, giving rise to what we term a "warm" posterior effect (WPE), i.e., there exist other posteriors ($p_\lambda$ with $\lambda < 1$) with lower testing loss, which, at the same time, have higher training loss due to Proposition 2. As a result, to describe CPE merely as a model misspecification issue without acknowledging underfitting offers a narrow interpretation of the problem.

The examples presented in Figure 2 help illustrate the results of Proposition 5 and provide concrete demonstrations of the theoretical insights discussed: when CPE shows up, tuning $\lambda$ is akin to finding another Bayesian posterior with a less misspecified likelihood and prior. However, we note that in this particular Bayesian linear regression setup, the new prior $q(\boldsymbol{\theta}|\boldsymbol{X}, \lambda)$ is always equal to initial prior $p(\boldsymbol{\theta})$ because the variance of the likelihood is assumed to be constant. Therefore, the analysis of the regression case in Section 4.2 does not directly apply here.

In the discussion regarding likelihood, we refer to the regression example in Section 4.1. Let's first have a look at Figure 2c, where the Gaussian likelihood model has a larger variance than the true data-generating process. By increasing $\lambda$, we obtain a likelihood model with a smaller variance (divided by $\lambda^2$, as shown in the regression example in Section 4.1), i.e., we induce a new likelihood with lower aleatoric uncertainty (Proposition 6). Such a new model is closer to the true data-generating distribution and less misspecified, thus enjoying better performance. The opposite can be seen in Figure 2b, where the Gaussian likelihood model has a lower variance than the true data-generating distribution and the WPE occurs.

## 5.3 The likelihood misspecification argument

Likelihood misspecification has also been identified as a cause of CPE, especially in cases where the dataset has been *curated* (Aitchison, 2021; Kapoor et al., 2022). Data curation often involves carefully selecting samples and labels to improve the quality of the dataset. As a result, the curated data-generating distribution typically presents very low aleatoric uncertainty, meaning that $\nu(\boldsymbol{y}|\boldsymbol{x})$ usually takes values very close to either 1 or 0. However, as previously discussed in (Aitchison, 2021; Kapoor et al., 2022), the standard likelihoods used in deep learning for image classification, like softmax or sigmoid, tend to allocate more spread-out probabilities to the outcomes, implicitly reflecting a higher level of aleatoric uncertainty. Therefore, their use in curated datasets that exhibit low uncertainty made them misspecified (Kapoor et al., 2022; Fortuin et al., 2022). To address this issue, alternative likelihood functions like the Noisy-Dirichlet model (Kapoor et al., 2022, Section 4) have been proposed, which better align with the characteristics of the curated data. On the other hand, introducing noise labels also alleviates the CPE, as demonstrated in Aitchison (2021, Figure 7). By introducing noise labels, we intentionally increase aleatoric uncertainty in the data-generating distribution, which aligns better with the high aleatoric uncertainty assumed by the standard Bayesian deep networks (Kapoor et al., 2022). Consequently, according to these works, the CPE can be strongly alleviated when the likelihood misspecification is addressed.

Our theoretical analysis aligns with these findings in Sections 4.1 and 4.4: fitting low aleatoric uncertainty data-generating distributions, e.g., $\nu(y|\boldsymbol{x}) \in \{0.01, 0.99\}$, with high aleatoric uncertainty likelihood functions e.g., $p(y|\boldsymbol{x}, \boldsymbol{\theta}) \in [0.2, 0.8]$, induces underfitting, and thus, CPE. The presence of underfitting is not mentioned at all by any of these previous works (Aitchison, 2021; Kapoor et al., 2022). On top of that, using Propositions 5 and 6, our work explains why the likelihood implicitly used by the tempered posterior with $\lambda > 1$ provides better generalizaton performance. Because, in this case, we are using a likelihood $q(\boldsymbol{\theta}|\boldsymbol{X}, \lambda)$ (Equation 8) with lower aleatoric uncertainty, which better aligns with the low aleatoric uncertainty data-generating distribution induced by curated datasets, thus reducing the degree of model misspecification.

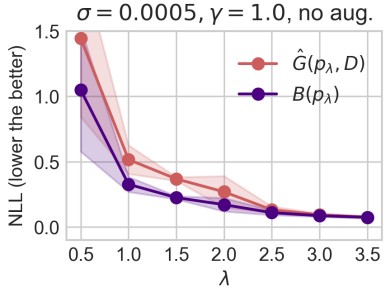 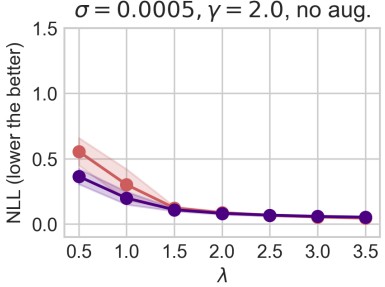 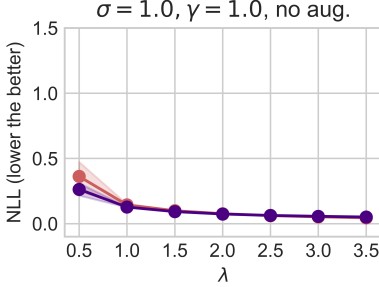

(a) Narrow prior and standard softmax     (b) Narrow prior and tempered softmax     (c) Standard prior and standard softmax

Figure 3: **Experimental illustrations for the arguments in Section 5 using small CNN via SGLD on MNIST. We show similar results on Fashion-MNIST with small CNN and CIFAR-10(0) with ResNet-18 in Appendix E**. Figures 3a to 3c illustrate the arguments in Section 5. Figure 3c uses the standard prior ($\sigma = 1$) and the standard softmax ($\gamma = 1$) for the likelihood without applying DA. Figure 3a follows a similar setup except for using a narrow prior. Figure 3b uses a narrow prior as in Figure 3a but with a tempered softmax that results in a lower aleatoric uncertainty. We report the training loss $\hat{G}(p_\lambda, D)$ and the testing losses, $B(p_\lambda)$ and $G(p_\lambda)$, from 10 samples of the small Convolutional neural network (CNN) via Stochastic Gradient Langevin Dynamics (SGLD). We show the mean and standard error across three different seeds. For additional experimental details, please refer to Appendix E.

Figures 3a and 3b, along with Figures 4a and 4b, illustrate this point through a regular multi-class classification task on a curated benchmark dataset. Both scenarios utilize the same narrow prior. The distinction in Figure 3b lies in the adoption of a tempered softmax likelihood, defined as $p(y|x, \theta) = (1 + \exp(-\gamma \operatorname{logits}(x, \theta)))^{-1}$, with $\gamma = 2$, compared to $\gamma = 1$ in Figure 3a. This tempered softmax likelihood, more closely aligned with the dataset's low aleatoric uncertainty as outlined by (Guo et al., 2017), leads to a reduced incidence of CPE in Figure 3b compared to Figure 3a. From the perspective of Proposition 5 and specifically Proposition 6, the intrinsic lower aleatoric uncertainty of the likelihood used in Figure 3b (softmax with $\gamma = 2$) makes

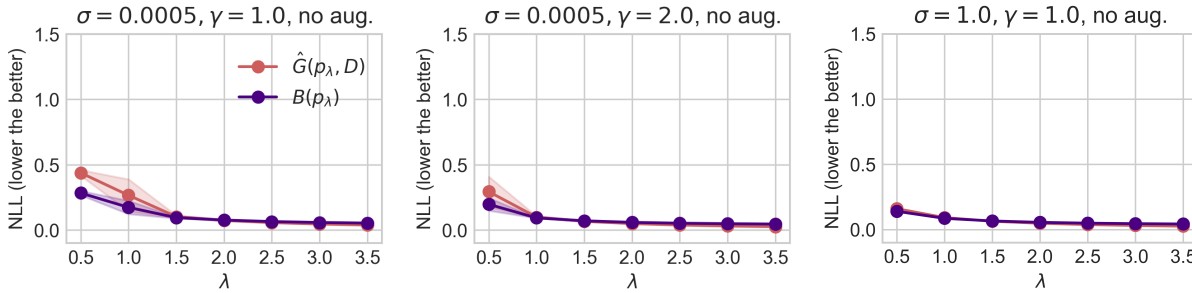

(a) Narrow prior and standard softmax    (b) Narrow prior and tempered softmax    (c) Standard prior and standard softmax

Figure 4: **Experimental illustrations for the arguments in Section 5 using large CNN via SGLD on MNIST. We show similar results on Fashion-MNIST with large CNN and CIFAR-10(0) with ResNet-50 in Appendix E.** The experiment setup is similar to the setups in Figure 3 but with a large CNN. Please refer to Appendix E for further details on the model.

the potential for improvement through increasing $\lambda$ somewhat limited, resulting in a less pronounced CPE compared to Figure 3a. It is, however, important to highlight the critical interaction between the likelihood and the prior, as we dicuss next.

### 5.4 The prior misspecification argument

As highlighted in previous works, such as in Wenzel et al. (2020); Fortuin et al. (2022), isotropic Gaussian priors are commonly chosen in modern Bayesian neural networks for the sake of tractability in approximate Bayesian inference rather than chosen based on their alignment with our actual beliefs. Given that the presence of the CPE implies that either the likelihood and/or the prior are misspecified, and given that neural networks define highly flexible likelihood functions, there are strong reasons for thinking these commonly used priors are misspecified. Notably, the experiments conducted by Fortuin et al. (2022) demonstrate that the CPE can be mitigated in fully connected neural networks when using heavy-tailed prior distributions that better capture the weight characteristics typically observed in such networks. However, such priors were found to be ineffective in addressing the CPE in convolutional neural networks (Fortuin et al., 2022), indicating the challenges involved in designing effective Bayesian priors within this context.

Our theoretical analysis provides a deeper insight into these observations. As discussed in Section 3, the absence of underfitting means the absence of CPE. This suggests flexible likelihood functions may still result in posteriors that underfit due to the prior's tendency to overly regularize. This may incur CPE when the strong prior fails to allocate enough probability to models that both fit the training data well and exhibit good generalization capabilities. As detailed in Section 4.2, employing tempered posteriors with $\lambda > 1$ effectively defines a conditional prior $q(\boldsymbol{\theta}|\boldsymbol{X}, \lambda)$ that favors models with lower aleatoric uncertainty. If such models align better with the training data compared to the original prior, then we may observe CPE. Also, the conditional prior $q(\boldsymbol{\theta}|\boldsymbol{X}, \lambda)$ with $\lambda > 1$ can be considered better specified than the original prior $p(\boldsymbol{\theta})$.

Figures 3a and 3c exemplify this situation. The prior in the case of Figures 3a is very narrow ($\sigma = 0.0005$), inducing strong regularization. Such a narrow prior results in a posterior that severely underfits the training data, evident from the high empirical Gibbs loss that deviates significantly from zero. Additionally, we observe a strong CPE. On the other hand, with a flatter prior in the case of Figure 3c, the CPE is considerably diminished. According to the discussion above and Sections 4.2 and 4.4, we know that the flatter prior allocates more probability mass to preferred models. Also, such preferred models have lower aleatoric uncertainty than the ones assigned initially by the narrow prior in the former case. To elaborate further, in the former case, the new prior $q(\boldsymbol{\theta}|\boldsymbol{X}, \lambda)$ with $\lambda > 1$ would place much more probability mass to models with lower aleatoric uncertainty than the narrow prior and strongly alleviating underfitting. In the second case, since the flatter prior already distributes probability mass more broadly across the model class, the room to shift probability mass to models with lower aleatoric uncertainty is more limited than that from a narrower prior, resulting in a milder CPE.

## 5.5 Model size, sample size in relation to CPE and underfitting

Larger models have the capacity to fit data more effectively, while smaller models are more likely to underfit. As we have argued that if there is no underfitting, there is no CPE, we expect that the size of the model has an impact on the strength of CPE as well. We demonstrate in Figure 3 and Figure 4. Specifically, in our experiments presented in Figure 3, we use a relatively small convolutional neural network (CNN), which has a more pronounced underfitting behavior, and this indeed corresponds to a stronger CPE. On the other hand, we employ a larger CNN in Figure 4, which has less underfitting, and we see the CPE is strongly alleviated. Actually, this effect can be directly inferred from Theorem 4. For an extremely flexible model capable of perfectly fitting both the original training samples and new samples, this theorem suggests that a CPE should not be expected, as the model's fit on the original data remains perfect, even when new examples are introduced.

Theorem 4 can also be used to understand why small models in the presence of large training data sets do not exhibit CPE. We empirically illustrate this point in Figure 2. In particular, Figure 2c and Figure 2e use the same (small) regression models and settings where the only difference is that Figure 2c uses 5 data points while Figure 2e uses 50 data points. In situations where a model possesses limited flexibility and the training set is large, including additional examples should barely affect the fit of the original training data because the Bayesian posterior is highly concentrated and will be barely affected by a single extra sample. Then, as predicted by Theorem 4, CPE in Figure 2e is much less significant than in Figure 2c.

Finally, it's worth noting that Figure 11 in Wenzel et al. (2020) shows the opposite effect, where larger models exhibit much stronger CPE compared to shallower or narrower versions of the same architectures. However, it's important to recognize that Wenzel et al. (2020) studied full-tempering, whereas our work focuses on likelihood-tempering. For full-tempering, Proposition 2 does not necessarily hold. Intuitively, since $\lambda$ operates on both the likelihood (data) and the prior (regularization) simultaneously, the effect of increasing $\lambda$ is mixed, not necessarily improving the fit on the training data. Consequently, the CPE brought by full-tempering as $\lambda$ increases does not necessarily coincide with better training loss, as the training loss may not be improvable. As a result, the CPE observed with full-tempering cannot be interpreted solely as underfitting. Therefore, for full-tempering, increasing model capacity may not achieve a lower degree of CPE, unlike the behavior we observed in our case focusing on likelihood-tempering.

## 6 Data augmentation (DA) and the CPE

> **Section Overview**
>
> We show conditions under which data augmentation exacerbates the CPE.
>
> - Section 6.1: starting with the Gibbs loss for clarity, we show that data augmentation induces a stronger CPE on the Gibbs loss if the augmented data provides more information about the data-generating process, increasing the correlation between the expected and empirical losses.
>
> - Section 6.2: extending the above idea, we show analogous conditions where data augmentation exacerbates CPE on the Bayes loss.

Machine learning is applied to many different fields and problems. In many of them, the data-generating distribution is known to have properties that can be exploited to generate new data samples (Shorten & Khoshgoftaar, 2019) artificially. This is commonly known as *data augmentation (DA)* and relies on the property that for a given set of transformations $H = \{h : \mathcal{X} \to \mathcal{X}\}$, the data-generating distribution satisfies $\nu(\boldsymbol{y}|\boldsymbol{x}) = \nu(\boldsymbol{y}|h(\boldsymbol{x}))$ for all $h \in H$. In practice, not all the transformations are applied to every single data. Instead, a probability distribution (usually uniform) $\mu_H$ is defined over $H$, and augmented samples are drawn accordingly. As argued in Nabarro et al. (2022), the use of data augmentation when training Bayesian neural networks implicitly targets the following (pseudo) log-likelihood, denoted $\hat{L}_{\mathrm{DA}}(D, \boldsymbol{\theta})$ and defined as

$$\hat{L}_{\mathrm{DA}}(D, \boldsymbol{\theta}) = \frac{1}{n} \sum_{i \in [n]} \mathbb{E}_{h \sim \mu_H} \left[ -\ln p(\boldsymbol{y}_i | h(\boldsymbol{x}_i), \boldsymbol{\theta}) \right] , \tag{12}$$

where data augmentation provides unbiased estimates of the expectation under the set of transformations using *Monte Carlo samples* (i.e., random data augmentations).

Although some argue that this data-augmented *(pseudo) log-likelihood* "does not have a clean interpretation as a valid likelihood function" (Wenzel et al., 2020; Izmailov et al., 2021), we do not need to enter into this discussion to understand why the CPE emerges when using the generalized Bayes posterior (Bissiri et al., 2016) associated to this *(pseudo) log-likelihood*, which is the main goal of this section. We call this posterior the DA-tempered posterior and is denoted by $p_\lambda^{\mathrm{DA}}(\boldsymbol{\theta}|D)$. The DA-tempered posterior can be expressed as the global minimizer of the following learning objective,

$$p_\lambda^{\mathrm{DA}}(\boldsymbol{\theta}|D) = \arg\min_\rho \mathbb{E}_\rho[n\hat{L}_{\mathrm{DA}}(D,\boldsymbol{\theta})] + \frac{1}{\lambda}\,\mathrm{KL}(\rho(\boldsymbol{\theta}|D), p(\boldsymbol{\theta}))\,. \tag{13}$$

This is similar to Equation 3 but now using $\hat{L}_{\mathrm{DA}}(D,\boldsymbol{\theta})$ instead of $\hat{L}(D,\boldsymbol{\theta})$, where we recall the notation $\hat{L}(D,\boldsymbol{\theta}) = -\frac{1}{n}\ln p(D|\boldsymbol{\theta})$. Hence, the resulting DA-tempered posterior is given by $p_\lambda^{\mathrm{DA}}(\boldsymbol{\theta}|D) \propto e^{-n\lambda\hat{L}_{\mathrm{DA}}(D,\boldsymbol{\theta})}p(\boldsymbol{\theta})$. In comparison, the tempered posterior $p_\lambda(\boldsymbol{\theta}|D)$ in Equation 2 can be similarly expressed as $e^{-n\lambda\hat{L}(D,\boldsymbol{\theta})}p(\boldsymbol{\theta})$.

There is large empirical evidence that DA induces a stronger CPE (Wenzel et al., 2020; Izmailov et al., 2021; Fortuin et al., 2022). Indeed, many of these studies show that if CPE is not present in our Bayesian learning settings, using DA makes it appear. According to our previous analysis, this means that the use of DA induces a stronger underfitting. To understand why this is case, we will take a step back and begin analyzing the impact of DA in the so-called Gibbs loss of the DA-Bayesian posterior $p_{\lambda=1}^{\mathrm{DA}}$ rather than the Bayes loss, as this will help us in understanding this puzzling phenomenon.

## 6.1 Data augmentation and CPE on the Gibbs loss

The expected Gibbs loss of a given posterior $\rho$, denoted $G(\rho)$, is a commonly used metric in the theoretical analysis of the *generalization performance* of Bayesian methods (Germain et al., 2016; Masegosa, 2020). The Gibbs loss represents the average of the expected log-loss of individual models under the posterior $\rho$, that is,

$$G(\rho) = \mathbb{E}_\rho[L(\boldsymbol{\theta})] = \mathbb{E}_\rho[\mathbb{E}_\nu[-\ln[p(\boldsymbol{y}|\boldsymbol{x},\boldsymbol{\theta})]]\,.$$

In fact, Jensen's inequality confirms that the expected Gibbs loss serves as an upper bound for the Bayes loss, i.e., $G(\rho) \geq B(\rho)$. This property supports the expected Gibbs loss to act as a proxy of the Bayes loss, which justifies its usage in gaining insights into how DA impacts the CPE.

We will now study whether data augmentation can cause a CPE on the Gibb loss. In other words, we will examine whether increasing the parameter $\lambda$ of the DA-tempered posterior leads to a reduction in the Gibbs loss. This can be formalized by extending Definition 1 to the expected Gibbs loss by considering its derivative $\frac{d}{d\lambda}G(p_\lambda)$ at $\lambda = 1$, which can be represented as follows:

$$\frac{d}{d\lambda}G(p_\lambda)_{|\lambda=1} = -\mathrm{COV}_{p_{\lambda=1}}\big(n\hat{L}(D,\boldsymbol{\theta}), L(\boldsymbol{\theta})\big)\,. \tag{14}$$

Where $\mathrm{COV}(X,Y)$ denotes the covariance of $X$ and $Y$. Again, due to the page limit, we postpone the necessary proofs in this section to Appendix C.

With this extended definition, if Equation 14 is negative, we can infer the presence of CPE for the Gibbs loss as well. Based on this, we say that DA induces a stronger CPE if the derivative of the expected Gibbs loss for the DA-tempered posterior exhibits a more negative trend at $\lambda = 1$, i.e., if $\frac{d}{d\lambda}G(p_\lambda^{\mathrm{DA}})_{|\lambda=1} < \frac{d}{d\lambda}G(p_\lambda)^{|\lambda=1}$. This condition can be equivalently stated as

$$\mathrm{COV}_{p_{\lambda=1}^{\mathrm{DA}}}\big(n\hat{L}_{\mathrm{DA}}(D,\boldsymbol{\theta}), L(\boldsymbol{\theta})\big) > \mathrm{COV}_{p_{\lambda=1}}\big(n\hat{L}(D,\boldsymbol{\theta}), L(\boldsymbol{\theta})\big) > 0\,. \tag{15}$$

The inequality presented above helps characterize and understand the occurrence of a stronger CPE when using DA. A stronger CPE arises if the expected Gibbs loss of a model $L(\boldsymbol{\theta})$ is more *correlated* with the empirical Gibbs loss of this model on the augmented training dataset $\hat{L}_{\mathrm{DA}}(D,\boldsymbol{\theta})$ than on the non-augmented

dataset $\hat{L}(D, \boldsymbol{\theta})$. This observation suggests that, if we empirically observe that the CPE is stronger when using an augmented dataset, the set of transformations $H$ used to generate the augmented dataset are introducing *valuable information* about the data-generating process.

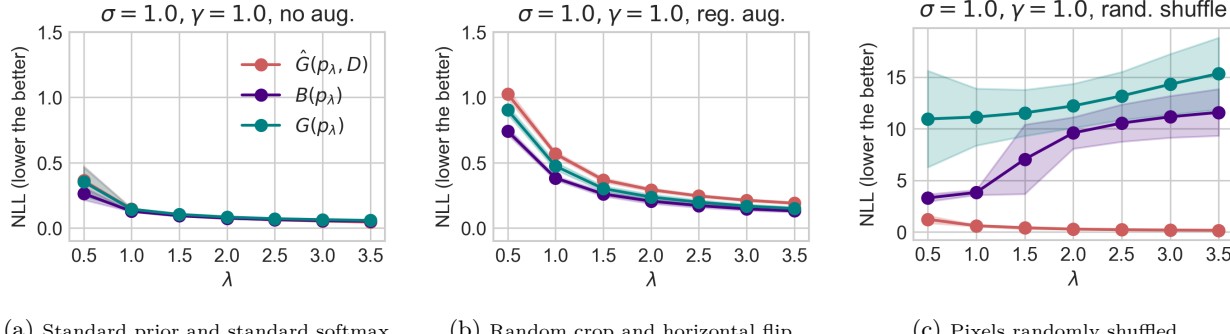

(a) Standard prior and standard softmax      (b) Random crop and horizontal flip      (c) Pixels randomly shuffled

Figure 5: **Experimental illustrations for the arguments in Section 6 using small CNN via SGLD on MNIST. We show similar results on Fashion-MNIST with small CNN and CIFAR-10(0) with ResNet-18 in Appendix E**. Figures 5a to 5c illustrate the arguments in Section 6. Figure 5a uses the standard prior ($\sigma = 1$) and the standard softmax ($\gamma = 1$) for the likelihood without applying DA. Figure 5b follows the setup as in Figure 5a but with standard DA applied, while Figure 5c uses fabricated DA. We report the training loss $\hat{G}(p_\lambda, D)$ and the testing losses $B(p_\lambda)$ and $G(p_\lambda)$ from 10 samples of the small Convolutional neural network (CNN) via Stochastic Gradient Langevin Dynamics (SGLD). We show the mean and standard error across three different seeds. For additional experimental details, please refer to Appendix E.

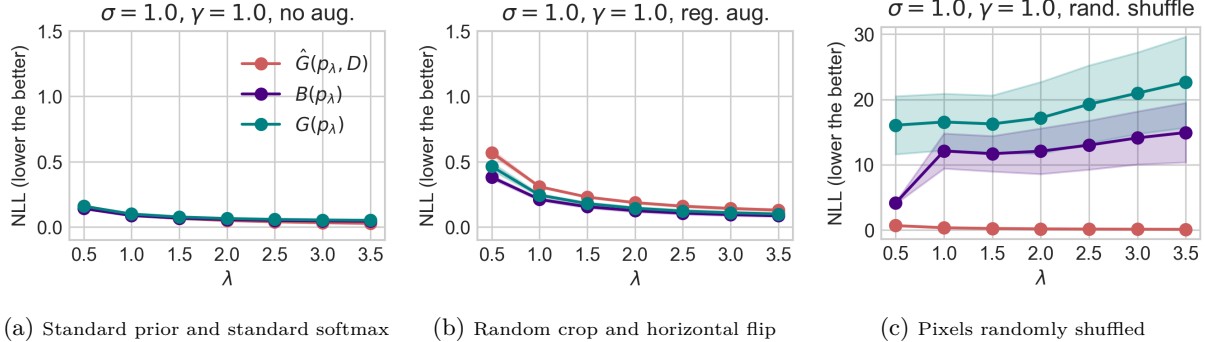

(a) Standard prior and standard softmax      (b) Random crop and horizontal flip      (c) Pixels randomly shuffled

Figure 6: **Experimental illustrations for the arguments in Section 6 using large CNN via SGLD on MNIST. We show similar results on Fashion-MNIST with large CNN and CIFAR-10(0) with ResNet-50 in Appendix E.** The experiment setup is similar to the setups in Figure 5 but with a large CNN. Please refer to Appendix E for further details on the model.

Figure 5 clearly illustrates such situations. Figure 5b shows that, compared to Figure 5a, the standard DA, which makes use of the invariances inherent in the data-generating distribution, induces a CPE on the Gibbs loss. Thus, the condition in Equation 15 holds by definition. On the other hand, Figure 5c uses a fabricated DA, where the same permutation is applied to the pixels of the images in the training dataset, which destroys low-level features present in the data-generating distribution. In this case, the derivative of the Gibb loss is positive, and Equation 15 holds in the opposite direction. These findings align perfectly with the explanations provided above, showing that DA induces a stronger underfitting.

## 6.2 Data augmentation and CPE on the Bayes loss

Now, we step aside of the Gibbs loss and focus back to the Bayes loss. The derivative of the Bayes loss at $\lambda = 1$ can also be written as,

$$\frac{d}{d\lambda} B(p_\lambda)_{|\lambda=1} = -\text{COV}_{p_{\lambda=1}} \left( n\hat{L}(D, \boldsymbol{\theta}), S_{p_{\lambda=1}}(\boldsymbol{\theta}) \right) , \tag{16}$$

where for any posterior $\rho$, $S_\rho(\boldsymbol{\theta})$ is a (negative) performance measure defined as

$$S_\rho(\boldsymbol{\theta}) = -\mathbb{E}_\nu \left[ \frac{p(\boldsymbol{y}|\boldsymbol{x}, \boldsymbol{\theta})}{\mathbb{E}_\rho[p(\boldsymbol{y}|\boldsymbol{x}, \boldsymbol{\theta})]} \right] . \tag{17}$$

This function measures the relative performance of a model parameterized by $\boldsymbol{\theta}$ compared to the average performance of the models weighted by $\rho$. Such measure is conducted on samples from the data-generating distribution $\nu(\boldsymbol{y}, \boldsymbol{x})$. Specifically, if the model $\boldsymbol{\theta}$ outperforms the average, we have $S_\rho(\boldsymbol{\theta}) < -1$, and if the model performs worse than the average, we have $S_p(\boldsymbol{\theta}) > -1$ (i.e., the lower the better). The derivations of the above equations are given in Appendix C.

By Definition 1 and Equation 16, DA will induce a stronger CPE if and only if the following condition is satisfied:

$$\text{COV}_{p_{\lambda=1}^{\text{DA}}} \left( n\hat{L}_{\text{DA}}(D, \boldsymbol{\theta}), S_{p_{\lambda=1}^{\text{DA}}}(\boldsymbol{\theta}) \right) > \text{COV}_{p_{\lambda=1}} \left( n\hat{L}(D, \boldsymbol{\theta}), S_{p_{\lambda=1}}(\boldsymbol{\theta}) \right) . \tag{18}$$

The previous analysis on the Gibbs loss remains applicable in this context, with the use of $S_\rho(\boldsymbol{\theta})$ as a metric for the expected performance on the true data-generating distribution instead of $L(\boldsymbol{\theta})$. While these metrics are slightly different, it is reasonable to assume that the same arguments we presented to explain the CPE under data augmentation for the Gibbs loss also apply here. The theoretical analysis aligns with the behavior of the Bayes loss as depicted in Figure 5.

Finally, comparing Figure 5b with Figure 6b, we also notice that using a larger neural network enables us to mitigate the CPE because we reduce the underfitting introduced by DA.

**Related work of the data augmentation argument.** The relation between data augmentation and CPE is an active topic of discussion (Wenzel et al., 2020; Izmailov et al., 2021; Noci et al., 2021; Nabarro et al., 2022). Some studies suggest that CPE is an artifact of DA because turning off data augmentation is enough to eliminate the CPE (Izmailov et al., 2021; Fortuin et al., 2022). Our study shows that this is *much more* than an artifact, as also argued in Nabarro et al. (2022). As discussed, the (pseudo) log-likelihood induced by standard DA is a better proxy of the expected log-loss, as precisely defined by Equation 15 and Equation 18.

Some argue that when using DA, we are not using a proper likelihood function (Izmailov et al., 2021), which could be a problem. Recent works (Nabarro et al., 2022) have developed principle likelihood functions that integrate DA-based approaches, hoping to remove CPE. However, they find that CPE still persists. Another widely accepted viewpoint regarding the interplay between the CPE and DA is that DA increases the effective sample size (Izmailov et al., 2021; Noci et al., 2021): "intuitively, data augmentation increases the amount of data observed by the model, and should lead to higher posterior contraction" (Izmailov et al., 2021).

Our analysis provides a more nuance understanding of this interplay between CPE and DA. First, we show that, when the augmented data provide extra information about the data-generating process, there is a stronger CPE, as shown in Equations 15 and 18. This, in turn, leads to higher posterior concentration. But, we also show that higher posterior concentration in the context of non-meaningful DA does not improve performance; as discussed before, Figure 5c illustrates this situation. Using the analysis given in Section 4, we can also add that tempering the posterior under DA is again a way to define alternative Bayesian posteriors that addresses this stronger underfitting, i.e., they better fit the training data and improve generalization.

## 7 Conclusions and limitations

Our research contributes to understanding the cold posterior effect (CPE) and its implications for Bayesian deep learning in several ways. Firstly, we theoretically demonstrate that the presence of the CPE implies that

the Bayesian posterior is underfitting. Secondly, by building on Zeno et al. (2021), we show that, in general, any tempered posterior can be considered as a proper Bayesian posterior with an alternative likelihood and prior distribution jointly parametrized by $T$, beyond merely the case of classification. Hence, fine-tuning the temperature parameter $T$ serves as an effective and theoretically sound approach to addressing the underfitting of the Bayesian posterior. Furthermore, we comprehensively discuss the interplay between several factors and CPE, including the use of approximate versus exact inference, model misspecification, and the size of the model and samples. Finally, our analysis in Section 6 reveals that data augmentation exacerbates the CPE by intensifying underfitting. This occurs because augmented data provides richer and more reliable information, enhancing the capacity for fitting.

Overall, our theoretical analysis underscores the significance of the CPE as an indicator of underfitting within the Bayesian framework and promotes the fine-tuning of the temperature $T$ in tempered posteriors as a principled approach to mitigate this issue. Furthermore, by dissecting the nature of the CPE and its effect on the Bayesian principle, our work aims to resolve ongoing debates and clarify the role of cold posteriors in enhancing the predictive performance of Bayesian deep learning models.

As a limitation of this work, we want to highlight that the characterization of CPE proposed here is defined only as the local change of Bayes loss at $\lambda = 1$. This approach does not account for scenarios where significant decreases in Bayes loss at other $\lambda$ values might also indicate the presence of CPE. We believe that our theoretical analysis could be expanded to include these cases as well.

## Acknowledgments

YW acknowledges support from Independent Research Fund Denmark, grant number 0135-00259B and Novo Nordisk Foundation, grant number NNF21OC0070621. The main part of the work was done when YW worked at University of Copenhagen. YZ acknowledge Ph.D. funding from Novo Nordisk A/S. AM acknowledges funding for cloud computing from Google Cloud for Researchers program and from the Junta de Andalucia, grant P20-00091, and UAL-FEDER, grant UAL2020-FQM-B1961. LO acknowledges financial support from project PID2022-139856NB-I00 funded by MCIN/ AEI / 10.13039/501100011033 / FEDER, UE and project PID2019-106827GB-I00 / AEI / 10.13039/501100011033 and from the Autonomous Community of Madrid (ELLIS Unit Madrid).

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

# A  Proofs for Section 3

In this section, we provide the proofs for Section 3 in the following order. We first prove the derivative of the empirical Gibbs loss in Proposition 2. Then, we show in Proposition 7 that for meaningful posteriors (depends on training data), the derivative won't be zero. Before proving Proposition 3 and Theorem 4, we first provide Proposition 8, stating an alternative expression of the derivative of the Bayes loss. The proofs of Proposition 3 and Theorem 4 then follow from that.

## A.1  Proof of Proposition 2

We first show a slightly more general result of $\frac{d}{d\lambda}\mathbb{E}_{p_\lambda}[f(\boldsymbol{\theta})]$ for any function $f(\boldsymbol{\theta})$ that is independent of $\lambda$. Recall that the posterior $p_\lambda(\boldsymbol{\theta}|D) \propto p(D|\boldsymbol{\theta})^\lambda p(\boldsymbol{\theta})$. With the fact that $\frac{d}{d\lambda}\left(p(D|\boldsymbol{\theta})^\lambda p(\boldsymbol{\theta})\right) = \ln(p(D|\boldsymbol{\theta}))p(D|\boldsymbol{\theta})^\lambda p(\boldsymbol{\theta})$, the derivative

$$\frac{d}{d\lambda}\mathbb{E}_{p_\lambda}[f(\boldsymbol{\theta})] = \mathbb{E}_{p_\lambda}[\ln p(D|\boldsymbol{\theta})f(\boldsymbol{\theta})] - \mathbb{E}_{p_\lambda}[\ln p(D|\boldsymbol{\theta})]\mathbb{E}_{p_\lambda}[f(\boldsymbol{\theta})] = \text{COV}_{p_\lambda}\left(\ln p(D|\boldsymbol{\theta}), f(\boldsymbol{\theta})\right), \tag{19}$$

where we denote $\text{COV}(X, Y)$ as the covariance of $X$ and $Y$. Hence, the derivative of the empirical Gibbs loss

$$\frac{d}{d\lambda}\hat{G}(p_\lambda, D) = \frac{d}{d\lambda}\mathbb{E}_{p_\lambda}[-\ln p(D|\boldsymbol{\theta})] = \text{COV}_{p_\lambda}\left(\ln p(D|\boldsymbol{\theta}), -\ln p(D|\boldsymbol{\theta})\right) = -\mathbb{V}_{p_\lambda}\left(\ln p(D|\boldsymbol{\theta})\right).$$

## A.2  Proposition 7

**Proposition 7.** *For any $\lambda > 0$ and $D \neq \emptyset$, if the tempered posterior $p_\lambda(\boldsymbol{\theta}|D) \propto p(D|\boldsymbol{\theta})^\lambda p(\boldsymbol{\theta})$ satisfies $\mathbb{V}_{p_\lambda}(\ln P(D|\boldsymbol{\theta})) = 0$, then, $p_\lambda(\boldsymbol{\theta}|D) = p(\boldsymbol{\theta})$.*

*Proof.* First of all, note that the tempered posterior is defined as

$$p_\lambda(\boldsymbol{\theta}|D) = \frac{p(D|\boldsymbol{\theta})^\lambda p(\boldsymbol{\theta})}{\int_{\boldsymbol{\theta}} p(D|\boldsymbol{\theta})^\lambda p(\boldsymbol{\theta})}.$$

Then,

$$\mathbb{V}_{p_\lambda}(\ln p(D|\boldsymbol{\theta})) = 0 \implies \int_{\boldsymbol{\theta}} p_\lambda(\boldsymbol{\theta}|D)\left(\ln p(D|\boldsymbol{\theta}) - \mathbb{E}_{p_\lambda}[\ln p(D|\boldsymbol{\theta})]\right)^2 = 0$$

Thus, for any $\boldsymbol{\theta} \in \text{supp}(p_\lambda)$, it verifies that

$$\ln p(D|\boldsymbol{\theta}) = \mathbb{E}_{p_\lambda}[\ln p(D|\boldsymbol{\theta})].$$

That is, $\ln p(D|\boldsymbol{\theta})$ is constant in the support of $p_\lambda$. Let $c$ denote such constant, then

$$p_\lambda(\boldsymbol{\theta}|D) = \frac{e^{c\lambda}p(\boldsymbol{\theta})}{\int_{\boldsymbol{\theta}} e^{c\lambda}p(\boldsymbol{\theta})} = \frac{e^{c\lambda}p(\boldsymbol{\theta})}{e^{c\lambda}\int_{\boldsymbol{\theta}} p(\boldsymbol{\theta})} = p(\boldsymbol{\theta}).$$

$\square$

## A.3  Proof of Proposition 3 and Theorem 4

In order to prove Proposition 3 and Theorem 4, we first show in Proposition 8 that the derivative of the Bayes loss of the tempered posterior $p_\lambda$ can be expressed by the difference between the empirical Gibbs loss of $\bar{p}_\lambda$ and the empirical Gibbs loss of $p_\lambda$.

**Proposition 8.** *The derivative of the Bayes loss of the tempered posterior $p_\lambda$ can be expressed by*

$$\frac{d}{d\lambda}B(p_\lambda) = \hat{G}(\bar{p}_\lambda, D) - \hat{G}(p_\lambda, D). \tag{20}$$

*Proof.* By definition,

$$\frac{d}{d\lambda}B(p_\lambda) = \frac{d}{d\lambda}\mathbb{E}_\nu[-\ln\mathbb{E}_{p_\lambda}[p(\boldsymbol{y}|\boldsymbol{x},\boldsymbol{\theta})]] = -\mathbb{E}_\nu\left[\frac{d}{d\lambda}\ln\mathbb{E}_{p_\lambda}[p(\boldsymbol{y}|\boldsymbol{x},\boldsymbol{\theta})]\right],$$

where

$$\frac{d}{d\lambda}\ln\mathbb{E}_{p_\lambda}[p(\boldsymbol{y}|\boldsymbol{x},\boldsymbol{\theta})] = \frac{\frac{d}{d\lambda}\mathbb{E}_{p_\lambda}[p(\boldsymbol{y}|\boldsymbol{x},\boldsymbol{\theta})]}{\mathbb{E}_{p_\lambda}[p(\boldsymbol{y}|\boldsymbol{x},\boldsymbol{\theta})]} = \frac{\mathrm{COV}_{p_\lambda}(\ln p(D|\boldsymbol{\theta}), p(\boldsymbol{y}|\boldsymbol{x},\boldsymbol{\theta}))}{\mathbb{E}_{p_\lambda}[p(\boldsymbol{y}|\boldsymbol{x},\boldsymbol{\theta})]}$$

due to Equation 19. By expanding the covariance, the above formula further equals to

$$\frac{\mathbb{E}_{p_\lambda}[\ln p(D|\boldsymbol{\theta})p(\boldsymbol{y}|\boldsymbol{x},\boldsymbol{\theta})] - \mathbb{E}_{p_\lambda}[\ln p(D|\boldsymbol{\theta})]\mathbb{E}_{p_\lambda}[p(\boldsymbol{y}|\boldsymbol{x},\boldsymbol{\theta})]}{\mathbb{E}_{p_\lambda}[p(\boldsymbol{y}|\boldsymbol{x},\boldsymbol{\theta})]} = \mathbb{E}_{\tilde{p}_\lambda}[\ln p(D|\boldsymbol{\theta})] - \mathbb{E}_{p_\lambda}[\ln p(D|\boldsymbol{\theta})],$$

where the probability distribution $\tilde{p}_\lambda(\boldsymbol{\theta}|D,(\boldsymbol{y},\boldsymbol{x})) \propto p_\lambda(\boldsymbol{\theta}|D)p(\boldsymbol{y}|\boldsymbol{x},\boldsymbol{\theta})$. Put everything together, we have

$$\frac{d}{d\lambda}B(p_\lambda) = \mathbb{E}_{p_\lambda}[\ln p(D|\boldsymbol{\theta})] - \mathbb{E}_\nu[\mathbb{E}_{\tilde{p}_\lambda}[\ln p(D|\boldsymbol{\theta})]] = \mathbb{E}_{p_\lambda}[\ln p(D|\boldsymbol{\theta})] - \mathbb{E}_{\bar{p}_\lambda}[\ln p(D|\boldsymbol{\theta})], \qquad (21)$$

where

$$\bar{p}_\lambda(\boldsymbol{\theta}|D) = \mathbb{E}_\nu[\tilde{p}_\lambda(\boldsymbol{\theta}|D,(\boldsymbol{y},\boldsymbol{x}))] = \mathbb{E}_\nu\left[\frac{p_\lambda(\boldsymbol{\theta}|D)p(\boldsymbol{y}|\boldsymbol{x},\boldsymbol{\theta})}{\mathbb{E}_{p_\lambda}[p(\boldsymbol{y}|\boldsymbol{x},\boldsymbol{\theta})]}\right].$$

The last equality is because

$$\begin{aligned}
\mathbb{E}_\nu[\mathbb{E}_{\tilde{p}_\lambda}[\ln p(D|\boldsymbol{\theta})]] &= \int_{(\boldsymbol{y},\boldsymbol{x})}\nu(\boldsymbol{y},\boldsymbol{x})\int_{\boldsymbol{\theta}}\tilde{p}_\lambda(\boldsymbol{\theta}|D,(\boldsymbol{y},\boldsymbol{x}))\ln p(D|\boldsymbol{\theta})\,d\boldsymbol{\theta}\,d(\boldsymbol{y},\boldsymbol{x}) \\
&= \int_{\boldsymbol{\theta}}\int_{(\boldsymbol{y},\boldsymbol{x})}\nu(\boldsymbol{y},\boldsymbol{x})\tilde{p}_\lambda(\boldsymbol{\theta}|D,(\boldsymbol{y},\boldsymbol{x}))\,d(\boldsymbol{y},\boldsymbol{x})\ln p(D|\boldsymbol{\theta})\,d\boldsymbol{\theta} \\
&= \int_{\boldsymbol{\theta}}\mathbb{E}_\nu[\tilde{p}_\lambda(\boldsymbol{\theta}|D,(\boldsymbol{y},\boldsymbol{x}))]\ln p(D|\boldsymbol{\theta})\,d\boldsymbol{\theta} \\
&= \mathbb{E}_{\bar{p}_\lambda}[\ln p(D|\boldsymbol{\theta})].
\end{aligned}$$

The last expression in Equation 21 further equals to $\hat{G}(\bar{p}_\lambda, D) - \hat{G}(p_\lambda, D)$ by definition. $\square$

### A.3.1 Proof of Proposition 3

Note that for any distribution $\rho$, we have $\hat{G}(\rho, D) := \mathbb{E}_\rho - \ln p(D|\boldsymbol{\theta}) \geq \min_{\boldsymbol{\theta}} - \ln p(D|\boldsymbol{\theta})$. On the other hand, Proposition 8 together with Definition 1 give that the CPE takes place if and only if

$$\frac{d}{d\lambda}B(p_\lambda)_{|\lambda=1} = \hat{G}(\bar{p}_{\lambda=1}, D) - \hat{G}(p_{\lambda=1}, D) < 0.$$

Therefore, it is not possible to have $\hat{G}(p_{\lambda=1}, D) \not> \min_{\boldsymbol{\theta}} - \ln p(D|\boldsymbol{\theta})$ and, at the same time, $\hat{G}(\bar{p}^{\lambda=1}, D) < \hat{G}(p_{\lambda=1}, D)$ because $\hat{G}(\bar{p}^{\lambda=1}, D) \geq \min_{\boldsymbol{\theta}} - \ln p(D|\boldsymbol{\theta})$.

### A.3.2 Proof of Theorem 4

It's easy to see from Proposition 8 that

$$\frac{d}{d\lambda}B(p_\lambda)_{|\lambda=1} = \hat{G}(\bar{p}_{\lambda=1}, D) - \hat{G}(p_{\lambda=1}, D) = 0$$

if and only if $\hat{G}(\bar{p}_{\lambda=1}, D) = \hat{G}(p_{\lambda=1}, D)$.

# B Proofs for Section 4

## B.1 Proof of Proposition 5

First of all, by the definition in Equation 2, and assuming a data-independent prior $p(\boldsymbol{\theta}|\boldsymbol{X}) = p(\boldsymbol{\theta})$, the tempered posterior is given by

$$p_\lambda(\boldsymbol{\theta}|\boldsymbol{X}, \boldsymbol{Y}) \propto p(\boldsymbol{Y}|\boldsymbol{X}, \boldsymbol{\theta})^\lambda p(\boldsymbol{\theta}),$$

where the tempered likelihood fully factorizes as $p(\boldsymbol{Y}|\boldsymbol{X}, \boldsymbol{\theta})^\lambda = \prod_{(\boldsymbol{y},\boldsymbol{x}) \in (\boldsymbol{Y}, \boldsymbol{X})} p(\boldsymbol{y}|\boldsymbol{x}, \boldsymbol{\theta})^\lambda$. Let a similar but $\boldsymbol{y}$-independent function $k(\boldsymbol{\theta}, \boldsymbol{X}, \lambda) = \prod_{\boldsymbol{x} \in \boldsymbol{X}} \int p(\boldsymbol{y}|\boldsymbol{x}, \boldsymbol{\theta})^\lambda \, d\boldsymbol{y}$.

Therefore, $p(\boldsymbol{Y}|\boldsymbol{X}, \boldsymbol{\theta})^\lambda p(\boldsymbol{\theta}) = \frac{p(\boldsymbol{Y}|\boldsymbol{X},\boldsymbol{\theta})^\lambda}{k(\boldsymbol{\theta},\boldsymbol{X},\lambda)} \left( k(\boldsymbol{\theta}, \boldsymbol{X}, \lambda) p(\boldsymbol{\theta}) \right)$, where we can let the new prior

$$q(\boldsymbol{\theta}|\boldsymbol{X}, \lambda) \propto p(\boldsymbol{\theta}) k(\boldsymbol{\theta}, \boldsymbol{X}, \lambda) = p(\boldsymbol{\theta}) \prod_{\boldsymbol{x} \in \boldsymbol{X}} \int p(\boldsymbol{y}|\boldsymbol{x}, \boldsymbol{\theta})^\lambda \, d\boldsymbol{y},$$

and the new posterior

$$q(\boldsymbol{Y}|\boldsymbol{X}, \boldsymbol{\theta}, \lambda) = \frac{p(\boldsymbol{Y}|\boldsymbol{X}, \boldsymbol{\theta})^\lambda}{k(\boldsymbol{\theta}, \boldsymbol{X}, \lambda)} = \frac{\prod_{(\boldsymbol{y},\boldsymbol{x}) \in (\boldsymbol{Y}, \boldsymbol{X})} p(\boldsymbol{y}|\boldsymbol{x}, \boldsymbol{\theta})^\lambda}{\prod_{\boldsymbol{x} \in \boldsymbol{X}} \int p(\boldsymbol{y}|\boldsymbol{x}, \boldsymbol{\theta})^\lambda \, d\boldsymbol{y}} = \prod_{(\boldsymbol{y},\boldsymbol{x}) \in (\boldsymbol{Y}, \boldsymbol{X})} q(\boldsymbol{y}|\boldsymbol{x}, \boldsymbol{\theta}).$$

## B.2 Proof of Proposition 6

The proof is made using differential entropy, i.e. assuming continuous target values $\boldsymbol{y}$. The only assumption is that Leibniz integral rule holds for $q(\boldsymbol{y}|\boldsymbol{x}, \boldsymbol{\theta}, \lambda) \ln q(\boldsymbol{y}|\boldsymbol{x}, \boldsymbol{\theta}, \lambda))$, verifying that

$$\frac{d}{d\lambda} \int (q(\boldsymbol{y}|\boldsymbol{x}, \boldsymbol{\theta}, \lambda) \ln q(\boldsymbol{y}|\boldsymbol{x}, \boldsymbol{\theta}, \lambda)) \, d\boldsymbol{y} = \int \frac{d}{d\lambda} (q(\boldsymbol{y}|\boldsymbol{x}, \boldsymbol{\theta}, \lambda) \ln q(\boldsymbol{y}|\boldsymbol{x}, \boldsymbol{\theta}, \lambda)) \, d\boldsymbol{y}.$$

In the case of supervised classification problems, we adopt the Shanon entropy, where equality holds naturally

$$\frac{d}{d\lambda} \sum_{\boldsymbol{y} \in \mathcal{Y}} (q(\boldsymbol{y}|\boldsymbol{x}, \boldsymbol{\theta}, \lambda) \ln q(\boldsymbol{y}|\boldsymbol{x}, \boldsymbol{\theta}, \lambda)) = \sum_{\boldsymbol{y} \in \mathcal{Y}} \frac{d}{d\lambda} (q(\boldsymbol{y}|\boldsymbol{x}, \boldsymbol{\theta}, \lambda) \ln q(\boldsymbol{y}|\boldsymbol{x}, \boldsymbol{\theta}, \lambda)).$$

From the definition of differential entropy, we got that

$$H(q(\boldsymbol{y}|\boldsymbol{x}, \boldsymbol{\theta}, \lambda)) = - \int q(\boldsymbol{y}|\boldsymbol{x}, \boldsymbol{\theta}, \lambda) \ln q(\boldsymbol{y}|\boldsymbol{x}, \boldsymbol{\theta}, \lambda) \, d\boldsymbol{y}.$$

Thus, taking derivative w.r.t. $\lambda$ and exchanging derivative and integral leads to the following expression

$$\frac{d}{d\lambda} H(q(\boldsymbol{y}|\boldsymbol{x}, \boldsymbol{\theta}, \lambda)) = - \int \frac{d}{d\lambda} (q(\boldsymbol{y}|\boldsymbol{x}, \boldsymbol{\theta}, \lambda) \ln q(\boldsymbol{y}|\boldsymbol{x}, \boldsymbol{\theta}, \lambda)) \, d\boldsymbol{y} = - \int (\ln q(\boldsymbol{y}|\boldsymbol{x}, \boldsymbol{\theta}, \lambda) + 1) \frac{d}{d\lambda} q(\boldsymbol{y}|\boldsymbol{x}, \boldsymbol{\theta}, \lambda) \, d\boldsymbol{y}.$$

Using that $\int \frac{d}{d\lambda} q(\boldsymbol{y}|\boldsymbol{x}, \boldsymbol{\theta}, \lambda) d\boldsymbol{y} = \frac{d}{d\lambda} \int q(\boldsymbol{y}|\boldsymbol{x}, \boldsymbol{\theta}, \lambda) d\boldsymbol{y} = 0$, simplifies the expression as

$$\frac{d}{d\lambda} H(q(\boldsymbol{y}|\boldsymbol{x}, \boldsymbol{\theta}, \lambda)) = - \int \ln q(\boldsymbol{y}|\boldsymbol{x}, \boldsymbol{\theta}, \lambda) \frac{d}{d\lambda} q(\boldsymbol{y}|\boldsymbol{x}, \boldsymbol{\theta}, \lambda) \, d\boldsymbol{y}.$$

Let us consider now the second term inside the integral. Using the derivative of the quotient rule leads to the following:

$$\frac{d}{d\lambda} q(\boldsymbol{y}|\boldsymbol{x}, \boldsymbol{\theta}, \lambda) = \frac{d}{d\lambda} \frac{p(\boldsymbol{y}|\boldsymbol{x}, \boldsymbol{\theta})^\lambda}{\int p(\boldsymbol{y}|\boldsymbol{x}, \boldsymbol{\theta})^\lambda \, d\boldsymbol{y}} = \frac{p(\boldsymbol{y}|\boldsymbol{x}, \boldsymbol{\theta})^\lambda \ln p(\boldsymbol{y}|\boldsymbol{x}, \boldsymbol{\theta})}{\int p(\boldsymbol{y}|\boldsymbol{x}, \boldsymbol{\theta})^\lambda \, d\boldsymbol{y}} - \frac{p(\boldsymbol{y}|\boldsymbol{x}, \boldsymbol{\theta})^\lambda \int p(\boldsymbol{y}|\boldsymbol{x}, \boldsymbol{\theta})^\lambda \ln p(\boldsymbol{y}|\boldsymbol{x}, \boldsymbol{\theta}) \, d\boldsymbol{y}}{(\int p(\boldsymbol{y}|\boldsymbol{x}, \boldsymbol{\theta})^\lambda \, d\boldsymbol{y})^2}.$$

Where, using the definition of $q(\boldsymbol{y}|\boldsymbol{x}, \boldsymbol{\theta}, \lambda)$, we got that

$$\frac{p(\boldsymbol{y}|\boldsymbol{x}, \boldsymbol{\theta})^\lambda \ln p(\boldsymbol{y}|\boldsymbol{x}, \boldsymbol{\theta})}{\int p(\boldsymbol{y}|\boldsymbol{x}, \boldsymbol{\theta})^\lambda \, d\boldsymbol{y}} = q(\boldsymbol{y}|\boldsymbol{x}, \boldsymbol{\theta}, \lambda) \ln p(\boldsymbol{y}|\boldsymbol{x}, \boldsymbol{\theta}),$$

and

$$\frac{p(\boldsymbol{y}|\boldsymbol{x},\boldsymbol{\theta})^\lambda \int p(\boldsymbol{y}|\boldsymbol{x},\boldsymbol{\theta})^\lambda \ln p(\boldsymbol{y}|\boldsymbol{x},\boldsymbol{\theta}) \, d\boldsymbol{y}}{(\int p(\boldsymbol{y}|\boldsymbol{x},\boldsymbol{\theta})^\lambda \, d\boldsymbol{y})^2} = q(\boldsymbol{y}|\boldsymbol{x},\boldsymbol{\theta},\lambda) \int q(\boldsymbol{y}|\boldsymbol{x},\boldsymbol{\theta},\lambda) \ln p(\boldsymbol{y}|\boldsymbol{x},\boldsymbol{\theta}) \, d\boldsymbol{y}$$

$$= q(\boldsymbol{y}|\boldsymbol{x},\boldsymbol{\theta},\lambda) \mathbb{E}_q[\ln p(\boldsymbol{y}|\boldsymbol{x},\boldsymbol{\theta})].$$

As a result, we got that

$$\int \ln q(\boldsymbol{y}|\boldsymbol{x},\boldsymbol{\theta},\lambda) \frac{d}{d\lambda} q(\boldsymbol{y}|\boldsymbol{x},\boldsymbol{\theta},\lambda) \, d\boldsymbol{y} = \mathbb{E}_q[\ln p(\boldsymbol{y}|\boldsymbol{x},\boldsymbol{\theta}) \ln q(\boldsymbol{y}|\boldsymbol{x},\boldsymbol{\theta},\lambda)] - \mathbb{E}_q[\ln q(\boldsymbol{y}|\boldsymbol{x},\boldsymbol{\theta},\lambda)]\mathbb{E}_q[\ln p(\boldsymbol{y}|\boldsymbol{x},\boldsymbol{\theta})]$$

Using $q(\boldsymbol{y}|\boldsymbol{x},\boldsymbol{\theta},\lambda)$ definition again:

$$\int \ln q(\boldsymbol{y}|\boldsymbol{x},\boldsymbol{\theta},\lambda) \frac{d}{d\lambda} q(\boldsymbol{y}|\boldsymbol{x},\boldsymbol{\theta},\lambda) \, d\boldsymbol{y} = \mathbb{E}_q[\ln p(\boldsymbol{y}|\boldsymbol{x},\boldsymbol{\theta}) \ln \frac{p(\boldsymbol{y}|\boldsymbol{x},\boldsymbol{\theta})^\lambda}{\int p(\boldsymbol{y}|\boldsymbol{x},\boldsymbol{\theta})^\lambda}] - \mathbb{E}_q[\ln \frac{p(\boldsymbol{y}|\boldsymbol{x},\boldsymbol{\theta})^\lambda}{\int p(\boldsymbol{y}|\boldsymbol{x},\boldsymbol{\theta})^\lambda}]\mathbb{E}_q[\ln p(\boldsymbol{y}|\boldsymbol{x},\boldsymbol{\theta})]$$

Where, expanding the logarithms the denominators cancel each other, leading to

$$\int \ln q(\boldsymbol{y}|\boldsymbol{x},\boldsymbol{\theta},\lambda) \frac{d}{d\lambda} q(\boldsymbol{y}|\boldsymbol{x},\boldsymbol{\theta},\lambda) \, d\boldsymbol{y} = \mathbb{E}_q[\ln p(\boldsymbol{y}|\boldsymbol{x},\boldsymbol{\theta}) \ln p(\boldsymbol{y}|\boldsymbol{x},\boldsymbol{\theta})^\lambda] - \mathbb{E}_q[\ln p(\boldsymbol{y}|\boldsymbol{x},\boldsymbol{\theta})^\lambda]\mathbb{E}_q[\ln p(\boldsymbol{y}|\boldsymbol{x},\boldsymbol{\theta})]$$

$$= \lambda \mathbb{V}(\ln p(\boldsymbol{y}|\boldsymbol{x},\boldsymbol{\theta})) \geq 0$$

As a result, the entropy is negative.

## C  Proofs for Section 6

### C.1  Proof of Equation 14

Note that

$$\frac{d}{d\lambda} G(p_\lambda) = \frac{d}{d\lambda} \mathbb{E}_{p_\lambda}[L(\boldsymbol{\theta})] = \mathrm{COV}_{p_\lambda}(\ln p(D|\boldsymbol{\theta}), L(\boldsymbol{\theta})) = \mathrm{COV}_{p_\lambda}(-\hat{L}(D,\boldsymbol{\theta}), L(\boldsymbol{\theta})),$$

where the second equality is by applying Equation 19. By taking $\lambda = 1$, we obtain the desired derivative.

### C.2  Proof of Equation 16

Recall from the proof of Theorem 8 that

$$\frac{d}{d\lambda} B(p_\lambda) = \mathbb{E}_{p_\lambda}[\ln p(D|\boldsymbol{\theta})] - \mathbb{E}_{\bar{p}_\lambda}[\ln p(D|\boldsymbol{\theta})] = \mathbb{E}_{\bar{p}_\lambda}[\hat{L}(D,\boldsymbol{\theta})] - \mathbb{E}_{p_\lambda}[\hat{L}(D,\boldsymbol{\theta})],$$

where $\bar{p}_\lambda(\boldsymbol{\theta}|D) = \mathbb{E}_\nu[\tilde{p}_\lambda(\boldsymbol{\theta}|D,(\boldsymbol{y},\boldsymbol{x}))]$ (Equation 6), and $\tilde{p}_\lambda(\boldsymbol{\theta}|D,(\boldsymbol{y},\boldsymbol{x})) \propto p_\lambda(\boldsymbol{\theta}|D)p(\boldsymbol{y}|\boldsymbol{x},\boldsymbol{\theta})$ is the distribution obtained by updating the posterior $p_\lambda$ with one new sample $(\boldsymbol{y},\boldsymbol{x})$.

Therefore,

$$\mathbb{E}_{\bar{p}_\lambda}[\hat{L}(D,\boldsymbol{\theta})] = \mathbb{E}_\nu \mathbb{E}_{\tilde{p}_\lambda}[\hat{L}(D,\boldsymbol{\theta})] = \mathbb{E}_\nu \left[ \mathbb{E}_{p_\lambda} \left[ \frac{p(\boldsymbol{y}|\boldsymbol{x},\boldsymbol{\theta})}{\mathbb{E}_{p_\lambda}[p(\boldsymbol{y}|\boldsymbol{x},\boldsymbol{\theta})]} \hat{L}(D,\boldsymbol{\theta}) \right] \right].$$

By Fubini's theorem, the above formula further equals to

$$\mathbb{E}_{p_\lambda} \left[ \mathbb{E}_\nu \left[ \frac{p(\boldsymbol{y}|\boldsymbol{x},\boldsymbol{\theta})}{\mathbb{E}_{p_\lambda}[p(\boldsymbol{y}|\boldsymbol{x},\boldsymbol{\theta})]} \hat{L}(D,\boldsymbol{\theta}) \right] \right] = \mathbb{E}_{p_\lambda} \left[ \mathbb{E}_\nu \left[ \frac{p(\boldsymbol{y}|\boldsymbol{x},\boldsymbol{\theta})}{\mathbb{E}_{p_\lambda}[p(\boldsymbol{y}|\boldsymbol{x},\boldsymbol{\theta})]} \right] \hat{L}(D,\boldsymbol{\theta}) \right] = \mathbb{E}_{p_\lambda} \left[ -S_{p_\lambda}(\boldsymbol{\theta}) \cdot \hat{L}(D,\boldsymbol{\theta}) \right].$$

On the other hand, since

$$\mathbb{E}_{p_\lambda}[-S_{p_\lambda}(\boldsymbol{\theta})] = \mathbb{E}_{p_\lambda} \left[ \mathbb{E}_\nu \left[ \frac{p(\boldsymbol{y}|\boldsymbol{x},\boldsymbol{\theta})}{\mathbb{E}_{p_\lambda}[p(\boldsymbol{y}|\boldsymbol{x},\boldsymbol{\theta})]} \right] \right] = \mathbb{E}_\nu \left[ \mathbb{E}_{p_\lambda} \left[ \frac{p(\boldsymbol{y}|\boldsymbol{x},\boldsymbol{\theta})}{\mathbb{E}_{p_\lambda}[p(\boldsymbol{y}|\boldsymbol{x},\boldsymbol{\theta})]} \right] \right] = 1,$$

we have

$$\mathbb{E}_{p_\lambda}[\hat{L}(D,\boldsymbol{\theta})] = \mathbb{E}_{p_\lambda}[\hat{L}(D,\boldsymbol{\theta})]\mathbb{E}_{p_\lambda}[-S_{p_\lambda}(\boldsymbol{\theta})].$$

By putting them altogether,

$$\frac{d}{d\lambda} B(p_\lambda) = \mathbb{E}_{p_\lambda} \left[ -S_{p_\lambda}(\boldsymbol{\theta}) \cdot \hat{L}(D,\boldsymbol{\theta}) \right] - \mathbb{E}_{p_\lambda}[\hat{L}(D,\boldsymbol{\theta})]\mathbb{E}_{p_\lambda}[-S_{p_\lambda}(\boldsymbol{\theta})] = -\mathrm{COV}\left(\hat{L}(D,\boldsymbol{\theta}), S_{p_\lambda}(\boldsymbol{\theta})\right).$$

## D    Experiment details for Bayesian linear regression on synthetic data with exact inference

In this section we detail the settings of the toy experiment using synthetic data and exact Bayesian linear regression in Figure 2. We also show extra results of the derivative of Gibbs loss and Bayes loss w.r.t to $\lambda$ approximated by samples.

To begin, we will outline the data-generating process for the synthetic data used in the experiment shown in Figure 2 and Figure 7. We sample $x$ uniformly from the $[-1, 1]$ interval and pass it through a Fourier transformation to construct the input of the data. That is, for a sampled $x$, the input $\boldsymbol{x}$ is constructed by a 10-dimensional Fourier basis function $\boldsymbol{\phi}(x) = [g_1(x), ..., g_K(x)]^T$ for $K = 10$, where the basis functions are defined as follows: $g_1(x) = \dfrac{1}{\sqrt{2\pi}}$, and for other odd values of $k$, $g_k(x) = \dfrac{1}{\sqrt{\pi}} \sin(kx)$, whereas for even values of $k$, $g_k(x) = \dfrac{1}{\sqrt{\pi}} \cos(kx)$. The distribution of the output $y \in \mathbb{R}$ given an input $\boldsymbol{x}$, denoted as $\nu(y|\boldsymbol{x})$, follows a Normal distribution with mean $\mathbf{1}^T \boldsymbol{x}$ and variance 1.0, where $\mathbf{1}$ is an all-ones vector. That is, $\nu(y|\boldsymbol{x}) = \mathcal{N}(\mathbf{1}^T \boldsymbol{x}, 1.0)$.

In our experiment, the likelihood model and the prior model are defined differently for the four settings in Figure 2. To enable exact inference, both the likelihood and the prior are Gaussian, which gives a closed-form solution for the posterior predictive. This choice also provides convenience when studying the CPE: different values of $\lambda$ on the likelihood term can be naturally absorbed into the Gaussian densities by adjusting the variance (dividing by $\lambda$) without hindering the exact inference step. We describe them in detail in the following.

1. No misspecification: likelihood $p(y|\boldsymbol{x}, \boldsymbol{\theta}) = \mathcal{N}(\boldsymbol{\theta}^T \boldsymbol{x}, 1.0)$, prior $p(\boldsymbol{\theta}) = \mathcal{N}(0, 2)$. This is the baseline for comparison.

2. Misspecified likelihood I: likelihood $p(y|\boldsymbol{x}, \boldsymbol{\theta}) = \mathcal{N}(\boldsymbol{\theta}^T \boldsymbol{x}, 0.15)$ (the order of Fourier transformation is $K = 20$, however note that it still contains the $K = 5$ data-generating process in its solution space), prior $p(\boldsymbol{\theta}) = \mathcal{N}(0, 2)$. In this case, the model is misspecified in a way that it has a smaller variance than the data-generating process.

3. Misspecified likelihood II: likelihood $p(y|\boldsymbol{x}, \boldsymbol{\theta}) = \mathcal{N}(\boldsymbol{\theta}^T \boldsymbol{x}, 3.0)$, prior $p(\boldsymbol{\theta}) = \mathcal{N}(0, 2)$. In this case, the model is misspecified in a way that it has a larger variance than the data-generating process. This is similar to one of the scenarios where CPE was found: the curated data has a lower aleatoric uncertainty than the model (Aitchison, 2021).

4. Misspecified prior: likelihood $p(y|\boldsymbol{x}, \boldsymbol{\theta}) = \mathcal{N}(\boldsymbol{\theta}^T \boldsymbol{x}, 1.0)$, prior $p(\boldsymbol{\theta}) = \mathcal{N}(0, 0.5)$. The prior is poorly specified in a way that it is tightly centered at 0 while the best $\boldsymbol{\theta}$ should be 1.

In all the experiments, every training set consists of only 5 samples. Since there are more parameters than the number of training data points, our setting falls within the "overparameterized" regime where CPE has been observed in Bayesian deep learning (Wenzel et al., 2020).

Continuing from Figure 2, where we show the Gibbs loss $\hat{G}(p_\lambda, D)$ (training) and the Bayes loss $B(p_\lambda)$ (testing) with respect to $\lambda$, we now show their derivatives $\frac{d}{d\lambda}\hat{G}(p_\lambda, D)$ (Equation 5) and $\frac{d}{d\lambda}B(p_\lambda)$ (Equation 20) respectively in Figure 7. Here the losses are included for a clearer depiction of the derivatives. To approximate the Bayes loss for generating the plot, we use 10000 data points sampled from the data-generating distribution. Also, the derivatives are approximated using 10000 samples from the exact posteriors. From Figure 7, we could clearly see that the derivatives perfectly characterize the losses in all four settings.

## E    Experiment details for Bayesian neural networks on image data with approximate inference

In this section, we first present in Appendix E.1 the architectures of the small and large CNNs used in this paper. As promised in the main text, we then provide results on additional image datasets trained with

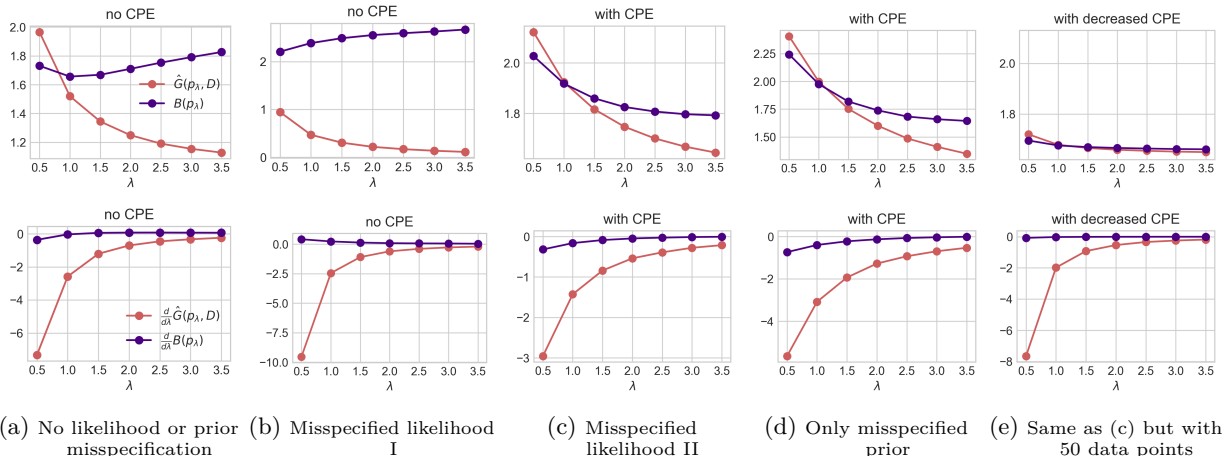

Figure 7: **The derivatives $\frac{d}{d\lambda}\hat{G}(p_\lambda, D)$ (Equation 5) and $\frac{d}{d\lambda}B(p_\lambda)$ (Equation 20) characterize the Gibbs loss $\hat{G}(p_\lambda, D)$ and the Bayes loss $B(p_\lambda)$ perfectly.**

Stochastic Gradient Langevin Dynamics (SGLD) (Welling & Teh, 2011) in Appendix E.2. Lastly, we provide additional results using mean-field variational inference (MFVI) (Blei et al., 2017) on MNIST, where we observe that the results of MFVI align with the ones with SGLD.

## E.1 Architectures of small/large CNN

**Small CNN** The small CNN is similar to LeNet-5, but with 107786 parameters in total:

1. Convolutional layer 1. Input channels: 1 (assuming grayscale images), output channels: 6, kernel size: 5x5, padding: 2, activation: ReLU.

2. Average pooling layer 1. Kernel size: 2x2, stride: 2.

3. Convolutional layer 2. Input channels: 6, output channels: 16, kernel size: 5x5, padding: 2, activation: ReLU.

4. Average pooling layer 2. Kernel size: 2x2, stride: 2.

5. Flattening layer. Flattens the output from the previous layers.

6. Fully connected layer 1. Input features: 784 (16 channels * 7 * 7), output features: 120, activation: ReLU.

7. Fully connected layer 2. Input features: 120, output features: 84, activation: ReLU.

8. Fully connected layer 3 (output layer). Input features: 84, output features: num_classes (specified during instantiation).

**Large CNN** The large CNN is similar to the small CNN, but with 545546 parameters in total:

1. Convolutional layer 1. Input channels: 1 (assuming grayscale images), output channels: 6, kernel size: 5x5, padding: 2, activation: ReLU.

2. Average pooling layer 1. Kernel size: 2x2, stride: 2.

3. Convolutional layer 2. Input channels: 6, output channels: 16, kernel size: 5x5, padding: 2, activation: ReLU.

4. Average pooling layer 2. Kernel size: 2x2, stride: 2.

5. Convolutional layer 3. Input channels: 16, output channels: 120, kernel size: 5x5, padding: 2, activation: ReLU.

6. Flattening layer. Flattens the output from the previous layers.

7. Fully connected layer 1. Input features: 5880 (120 channels $\times$ 7 $\times$ 7), output features: 84, activation: ReLU.

8. Fully connected layer 2 (output layer). Input features: 84, output features: num_classes (specified during instantiation).

In all the convolutional layers, no stride $= 1$ and padding is set to *same*.

### E.2 Stochastic Gradient Langevin Dynamics (SGLD)

Our experiments using SGLD are categorized into 4 groups:

1. Bayesian CNNs (small and large) on MNIST (Figures 3 - 6 in the main text)

2. Bayesian CNNs (small and large) on Fashion-MNIST (Appendix E.2.1)

3. Bayesian ResNets (18 and 50) on CIFAR-10 (Appendix E.2.2)

4. Bayesian ResNets (18 and 50) on CIFAR-100 (Appendix E.2.3)

where each group evaluates the effect of underfitting on a small model and a large model. Note that as we follow the standard ResNet-18 and ResNet-50, the details of the architectures are omitted. They have around 11 million and 23 million parameters, respectively. We implement with PyTorch (Paszke et al., 2019) and train the model using cyclical learning rate SGLD (cSGLD) (Zhang et al., 2019) for 1000 epochs. We set the learning rate to 1e-6 with a momentum term of 0.99. We run cSGLD for 10 trials and collect 10 samples for each trial. Experiments were conducted on NVIDIA A100 GPU, with each trial taking around 30 hours.

### E.2.1 Small and Large CNNs via SGLD on Fashion-MNIST

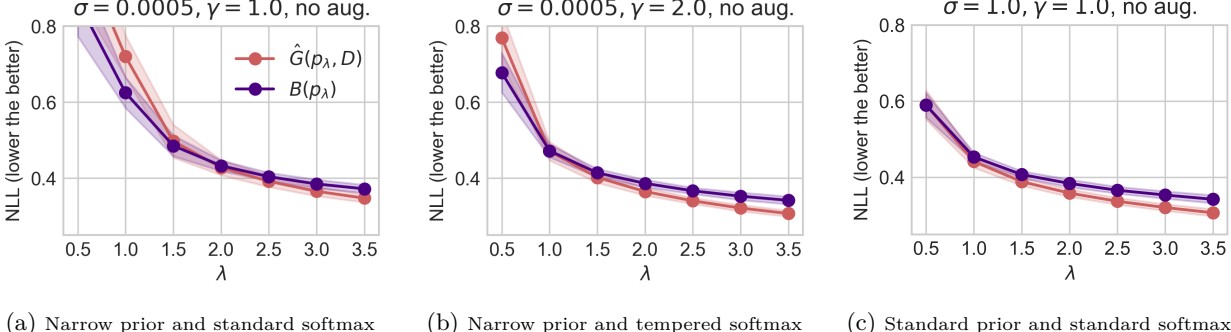

(a) Narrow prior and standard softmax  (b) Narrow prior and tempered softmax  (c) Standard prior and standard softmax

Figure 8: Extended results of Figure 3 using small CNN via SGLD on Fashion-MNIST.

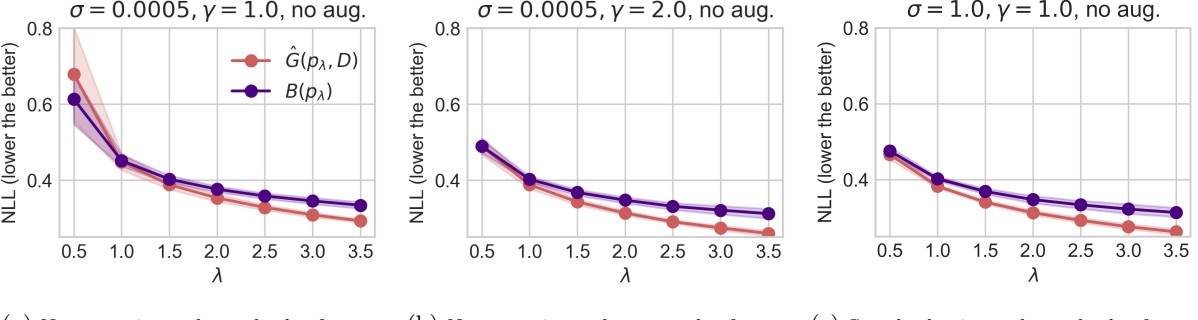

(a) Narrow prior and standard softmax  (b) Narrow prior and tempered softmax  (c) Standard prior and standard softmax

Figure 9: Extended results of Figure 4 using large CNN via SGLD on Fashion-MNIST.

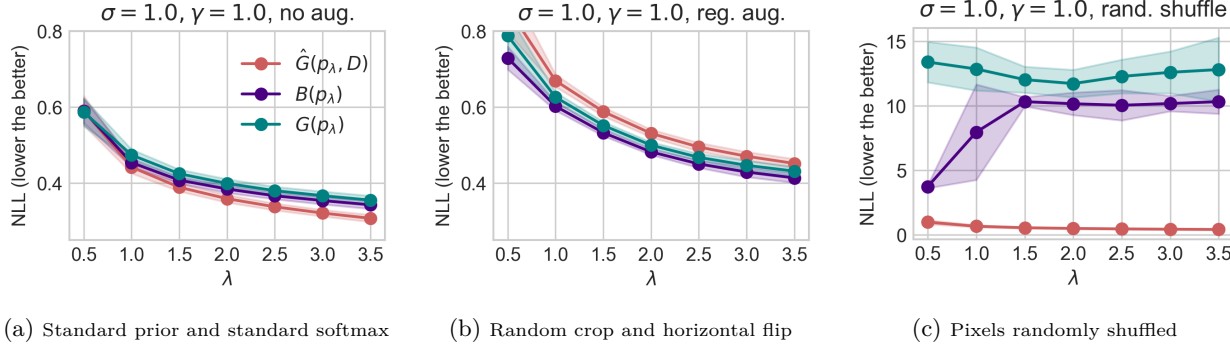

(a) Standard prior and standard softmax     (b) Random crop and horizontal flip     (c) Pixels randomly shuffled

Figure 10: Extended results of Figure 5 using small CNN via SGLD on Fashion-MNIST.

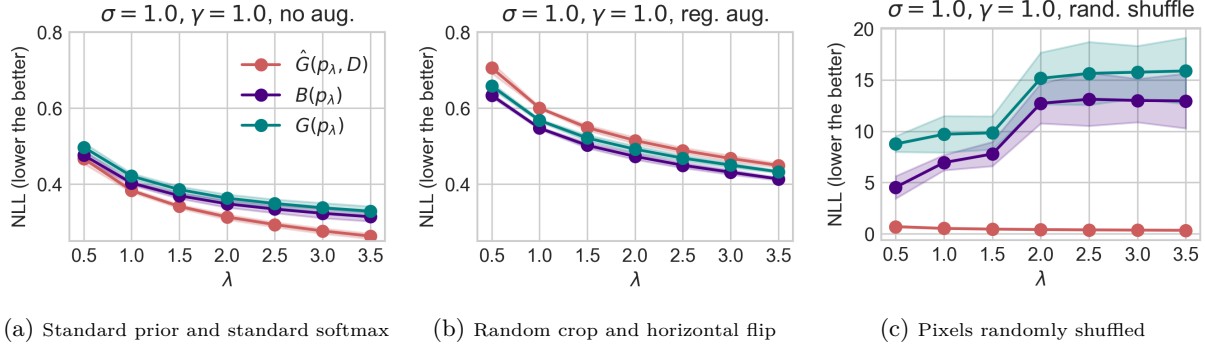

(a) Standard prior and standard softmax     (b) Random crop and horizontal flip     (c) Pixels randomly shuffled

Figure 11: Extended results of Figure 6 using large CNN via SGLD on Fashion-MNIST.

### E.2.2 ResNet-18 and ResNet-50 via SGLD on CIFAR-10

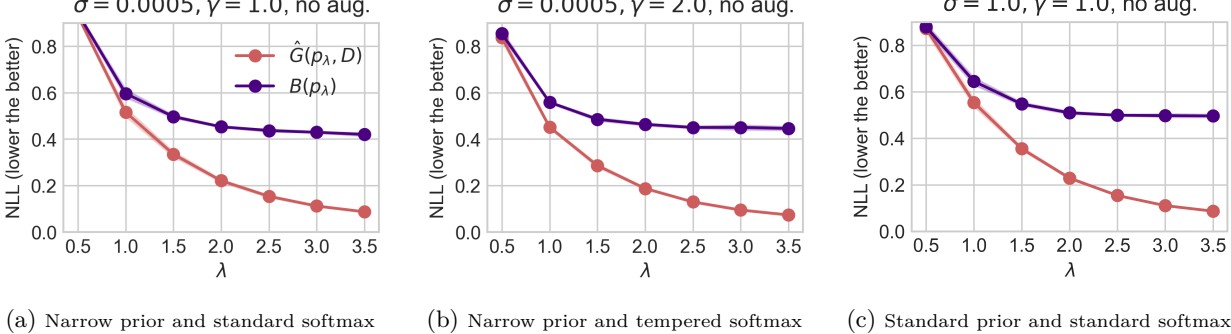

(a) Narrow prior and standard softmax  (b) Narrow prior and tempered softmax  (c) Standard prior and standard softmax

Figure 12: Extended results of Figure 3 using ResNet-18 via SGLD on CIFAR-10.

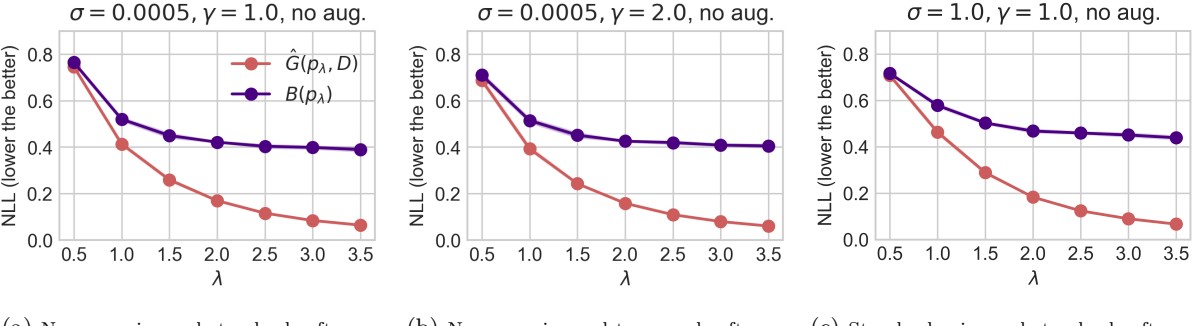

(a) Narrow prior and standard softmax  (b) Narrow prior and tempered softmax  (c) Standard prior and standard softmax

Figure 13: Extended results of Figure 4 using ResNet-50 via SGLD on CIFAR-10.

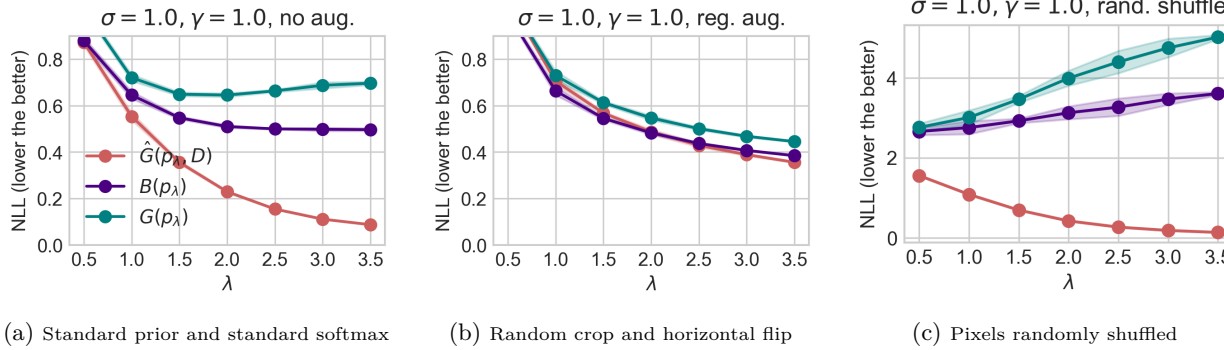

(a) Standard prior and standard softmax    (b) Random crop and horizontal flip    (c) Pixels randomly shuffled

Figure 14: Extended results of Figure 5 using ResNet-18 via SGLD on CIFAR-10.

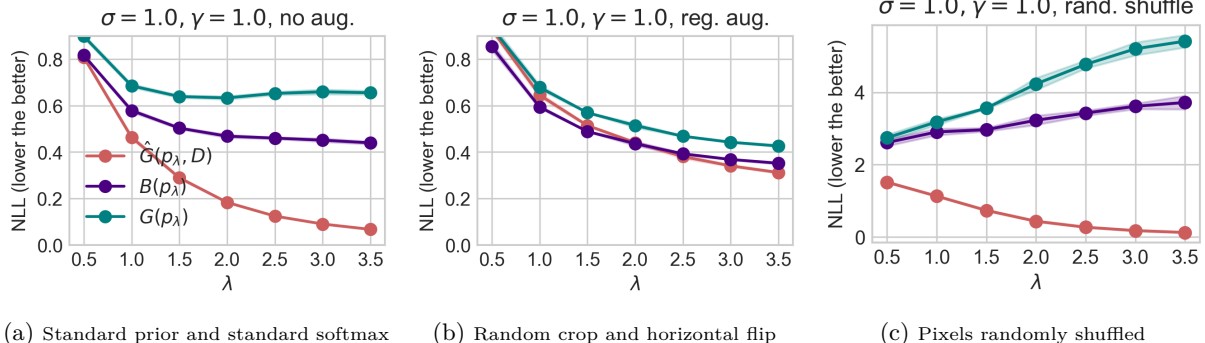

(a) Standard prior and standard softmax    (b) Random crop and horizontal flip    (c) Pixels randomly shuffled

Figure 15: Extended results of Figure 6 using ResNet-50 via SGLD on CIFAR-10.

### E.2.3 ResNet-18 and ResNet-50 via SGLD on CIFAR-100

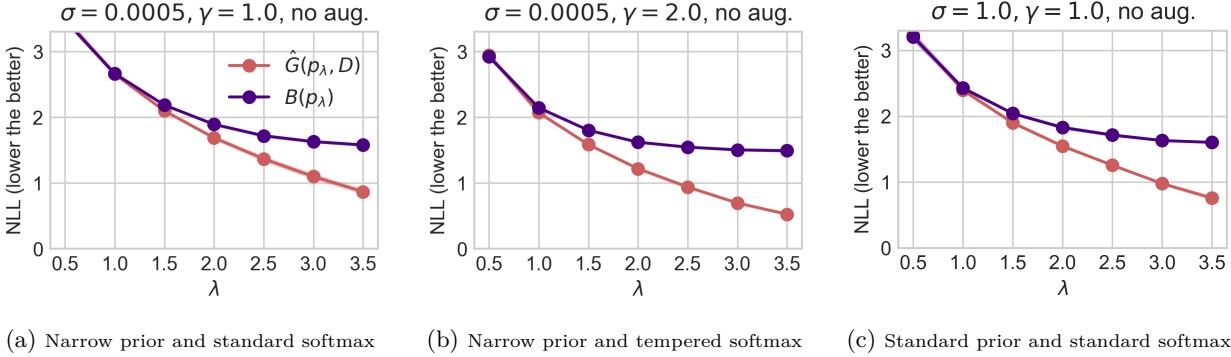

(a) Narrow prior and standard softmax    (b) Narrow prior and tempered softmax    (c) Standard prior and standard softmax

Figure 16: Extended results of Figure 3 using ResNet-18 via SGLD on CIFAR-100.

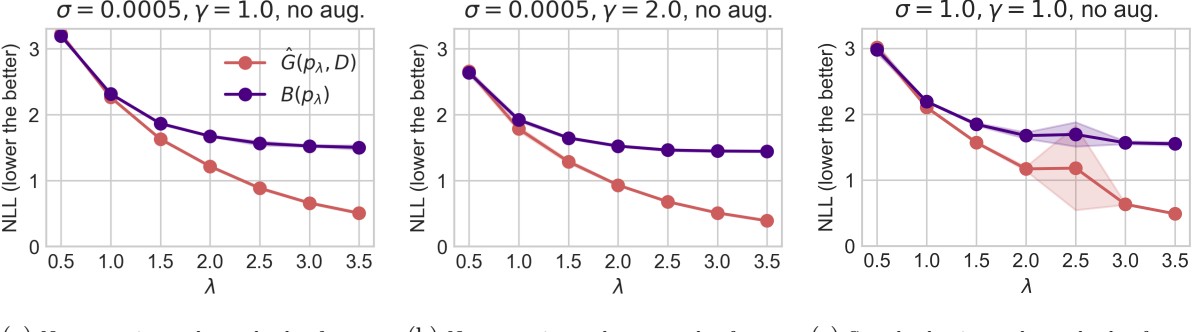

(a) Narrow prior and standard softmax    (b) Narrow prior and tempered softmax    (c) Standard prior and standard softmax

Figure 17: Extended results of Figure 4 using ResNet-50 via SGLD on CIFAR-100.

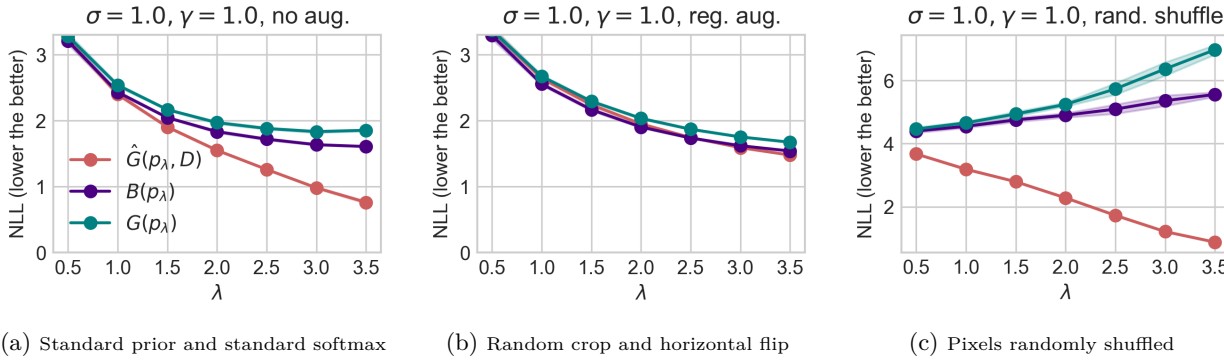

(a) Standard prior and standard softmax     (b) Random crop and horizontal flip     (c) Pixels randomly shuffled

Figure 18: Extended results of Figure 5 using ResNet-18 via SGLD on CIFAR-100.

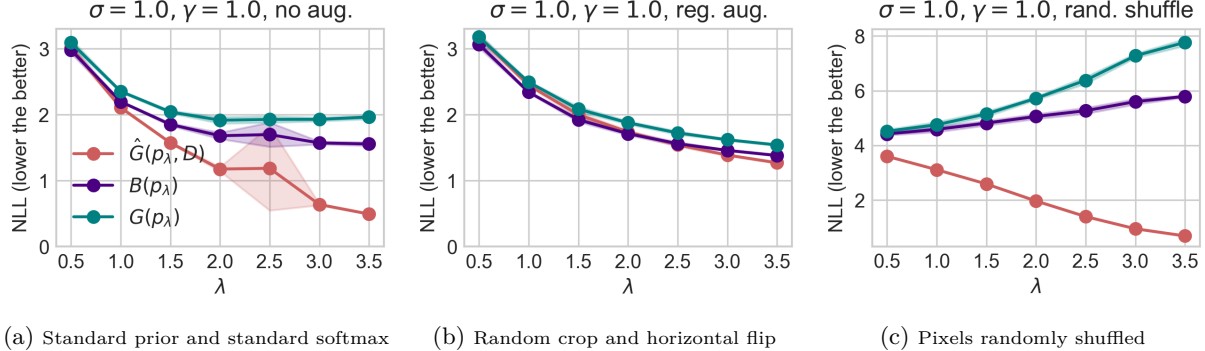

(a) Standard prior and standard softmax     (b) Random crop and horizontal flip     (c) Pixels randomly shuffled

Figure 19: Extended results of Figure 6 using ResNet-50 via SGLD on CIFAR-100.

### E.3 Mean-Field Variational Inference (MFVI)

**Experimental Settings:** These experiments were run using Tensorflow (Abadi et al., 2015), Tensorflow Probability (Dillon et al., 2017) and Keras (Chollet et al., 2015). By default, we use zero-center Normal distributions, $\mathcal{N}(0, \sigma)$, as priors with different standard deviations, i.e., $\sigma$ values. For the variational approximation, we use fully factorized Normal distributions, where both the mean and the standard deviation of each of them were the parameters to be learned by the variational algorithm. Although using an over-simplified family to approximate the true posterior, MFVI also achieves competitive results (Zhang & Nalisnick, 2021) compared to SGLD.

The convolutional neural network used for this experiment is a variational implementation of the network described above. This variational model uses a total of 1091092 parameters, double the number of parameters of the original model.

We use an Adam optimizer with a default learning rate 0.001, batch size = 100, and run during 100 epochs, which in our case, is enough to achieve convergence. The Keras global seed was set to 15. Other seeds were set, but similar results were obtained. Experiments were performed on Google Colab on a NVIDIA T4 GPU. The computation time was in the order of a few hours.

**Prior Misspecification, Likelihood Misspecification and the CPE:**

We run a similar experiment to the one reported in Figure 4 but using MFVI (Blei et al., 2017) as an approximate inference technique. The results of this experiment are reported in Figure 20. The conclusions are completely similar to the ones already discussed in Section 5.

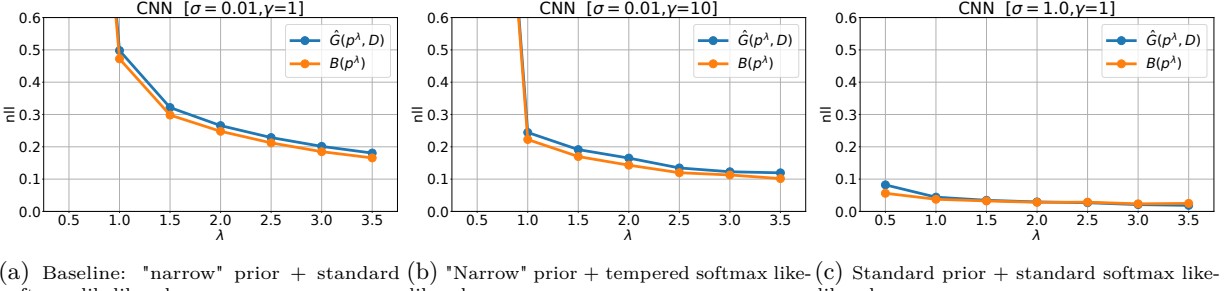

(a) Baseline: "narrow" prior + standard softmax likelihood

(b) "Narrow" prior + tempered softmax likelihood

(c) Standard prior + standard softmax likelihood

Figure 20: **CPE can be mitigated by a less misspecified model (Figure 20b) or imposing a less regularizing prior (Figure 20c).** We plot the training loss $\hat{G}(p_\lambda, D)$ and the testing loss $B(p_\lambda)$ with different priors and likelihood models. The parameter $\sigma$ is the standard deviation of the isotropic Gaussian prior centered at zero, while the parameter $\gamma$ serves as a smoothing parameter on the logits. All metrics are approximated using 10 samples drawn from the MFVI posterior.

**Data Augmentation (DA) and the CPE:**

As in the previous case, we ran a similar experiment to the one reported in Figure 4 but using MFVI (Blei et al., 2017) as an approximate inference technique. The results of this experiment are reported in Figure 21. The conclusions are very similar to the ones already discussed in Section 6.

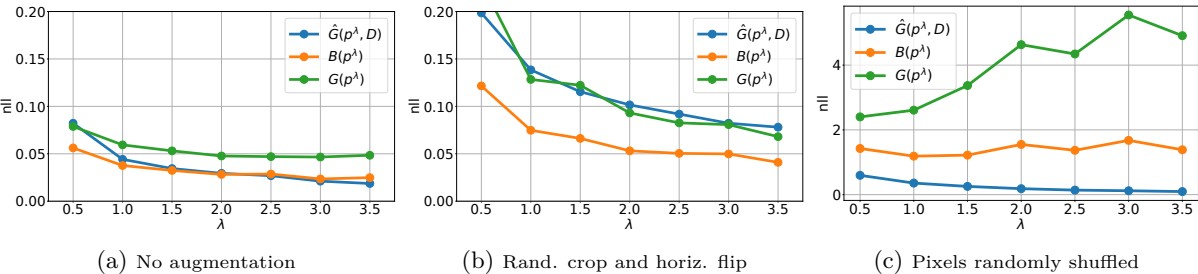

(a) No augmentation      (b) Rand. crop and horiz. flip      (c) Pixels randomly shuffled

Figure 21: **CPE only occurs with "meaningful" augmentation (Figure 21b).** We plot the training loss $\hat{G}(p_\lambda, D)$ and the testing losses $B(p_\lambda)$ and $G(p_\lambda)$ with different augmentation methods. While Figure 20 shows no augmentation, Figure 21b and 21c show standard augmentation and an artificially designed "harmful" augmentation, where the pixels are shuffled randomly. All metrics are approximated using 10 samples drawn from the MFVI posterior.

