# OpenReview forum: "The Cold Posterior Effect Indicates Underfitting, and Cold Posteriors Represent a Fully Bayesian Method to Mitigate It"
_TMLR — Accepted by TMLR_

### Review · Reviewer_GSTQ · 2024-04-22

**Summary Of Contributions:**

The authors propose an explanation for the Cold Posterior Effect (CPE). Their theory states that the CPE can be understood in terms of the Bayes posterior ($T=1$) underfitting, which can be alleviated by likelihood tempering ($T<1$). They show that certain misspecification of the prior and/or likelihood can result in such underfitting (at $T=1$) and mitigated by tempering, and discuss in depth how this relates to previous hypotheses of the CPE (Aitchison, 2021; Fortuin et al., 2022; Noci et al., 2021; Kapoor et al., 2022; Wenzel et al., 2020). Further, they demonstrate their theory predicts a more pronounced CPE when training with data augmentation (DA) whose transforms are consistent with invariances in the data. This prediction is consistent with the findings of many previous works which have investigated the relation between DA and the CPE (Izmailov et al., 2021; Nabarro et al., 2022; Kapoor et al., 2022).

**Audience:**

Yes

**Claims And Evidence:**

Yes

**Requested Changes:**

# Critical
1. Include references to Adlam et al., 2020 and Zeno et al., 2020, and revise the claims made in the current submission relative to these previous works.

2. Either justify why the current approach can be viewed as Empicial Bayes, or remove reference to EB.

# Minor
1. For completeness, include test accuracies in the image classification results.
2. Experimentally validate Theorem 4 by characterising how the strength of the CPE varies with the size of training set.
3. Show results for the standard prior + tempered softmax ($\sigma=1,\gamma=2$) case in Figs 3 and 4.

## Very Minor
- $p^\lambda(\mathbf{\theta}|D)$ would be clearer as $p_\lambda (\mathbf{\theta}|D)$ so as not to be confused with "to the power of $\lambda$​"
- Section 2: "If it induces no ambiguity" -> "as it induces no ambiguity"
- Section 3: Should eqn (5) be "Proposition **1**" rather than "Proposition 2"? I do not see a Proposition 1 anywhere in the paper.
- Section 3: "We will then show that the tempered posterior..." -> "We will **now** show that the tempered posterior..."
- Square brackets for outer expectation over $\nu$ missing in proof pof proposition 4 (bottom line of pg 17) and mid way down pg 18
- Section 4 repetition: "We also show that higher $\lambda$ values result in likelihood distributions with lower aleatoric uncertainty". This is already mentioned before eqn (9).
- Section 4: "...using the tempered posteriors implies implicitly using the prior..." -> "...using the tempered posteriors implies use of the prior..."
- Section 6: Do not denote DA transformations with $T$ which has already been defined as temperature.
- The mathematical results in Section 6 are satisfying and intuitive: there will be a larger CPE if the loss function used for inferring parameters is more strongly correlated with the expected Gibbs loss (or $S(\theta)$). The authors show that (for some datasets) the loss which incorporates DA is more strongly correlated to $L(\theta)$ than that which does not, therefore predicting a stronger CPE when using DA. However, I believe the reader would benefit from an intuitive explanation as to how this relates to the overall argument that underfitting causes a CPE.

**Strengths And Weaknesses:**

# Strengths
1. The central argument (that underfitting may be the main cause of the CPE) is a compelling one which is valuable to the ongoing discussion around the CPE.

2. The paper is relatively rigorous. The authors' attempt to formalise the notion of the CPE allows them to argue their case from a mathematical perspective as well as an empirical one. The latter has been the basis of most previous CPE-related works.

3. The paper is well-structured and, in general, well written. The narrative is coherent throughout.

4. Toy experiments (Fig 2) neatly demonstrate the authors' argument, in addition to the standard image classification experiments (MNIST, Fashion-MNIST, CIFAR-10, CIFAR-100).

5. For the most part, the authors do a good job of contextualising their work within the existing CPE-related literature. Particularly in Sections 5 and 6, the relation between their hypothesis and those proposed previously is clear.


# Weaknesses
1. While the majority of relevant work is properly cited, two important references are missing — Adlam et al., 2020 and Zeno et al., 2020 (see full references below). Adlam et al., 2020 show that i) cold posteriors can be caused by priors which overestimate aleatoric uncertainty, ii) for Gaussian processes the tempered posterior corresponds to choosing a different prior, which can be viewed as empirical Bayes. They do not link these observations to underfitting, but clearly this is related work — their Fig 3 illustrates a similar finding to Fig 2 in the authors' work.

    I believe Zeno et al., 2020 is yet more pertinent. They show that tempered posteriors can be viewed as the Bayes posteriors for different, but valid models, and derive the forms of the modified prior and likelihood. They also discuss how these modified priors encourage confident predictions at the training inputs. As far as I can tell, this negates the novelty of Contribution ii) of this paper and much of Section 4. Again, Zeno et al., 2020 do not view this through the lens of underfitting, so I believe the submitted work is still valuable, but needs to be properly qualified relative to Zeno et al., 2020.

2. The paper recognises that $T$ may be viewed as a model hyperparameter and tuned via Empirical Bayes (EB). I agree, and believe this is a worthwhile research direction. However, it is not clear to me that the method of tuning $T$ proposed in the paper is a form of EB. In particular, EB describes the hyperparameters wrt the _marginal likelihood_ of the _training data_ given the model (after parameters have been marginalised out). I therefore do not see how choosing $T$ based on the performance of a held-out validation set can be viewed as EB. I am happy to be proven wrong on this point.

## Minor Weaknesses
1. I definitely agree that eqn (4) is not the best possible characterisation of the CPE. In particular it only reflects the _local_ gradient of performance at $T=1$, where one may find e.g. performance is flat wrt $T$ at $T=1$, but perfomance improves as $T$ is reduced from $0.1$ to $0.01$. I believe this would still be intuitively viewed as a CPE.

2. The argument in Section 3 claims that $T=1$ should be optimal when additional examples from the data generating distribution do not improve the model fit on the original training data. This is an interesting result, but the implications should be discussed in more detail.

    i) Are the following predictions correct? If so I believe they should be discussed in the paper.

        A. For a very flexible model which can perfectly fit the original training set and new examples, we should not expect a CPE as the training fit on the original data will still be perfect even with the new examples.

        B. For a model with little flexibility and a small training set, the additional examples should reduce the impact of noise in the original training data and therefore improve the model fit on the original training data. We should expect a CPE in this case.

        C. For a model with little flexibility but infinite training data, the additional examples should not affect the model fit on the original data. We should not expect a CPE.

    ii) I believe A is demonstrated experimentally by comparing Figs 3 and 4. However, I would like to see B and C verified empirically, i.e. show that the CPE is more pronounced for smaller training sets with the same model.


## Clarification
I would be grateful if the authors could clarify the following point in relation to their central argument. The gist of the argument is:

1. Underfitting is is defined as the scenario in which "the trained model exhibits (much) higher training and testing losses compared to what is achievable".

2. Due to how tempering is defined (applied to the likelihood only), increasing $\lambda$ is guaranteed to reduce the training loss of a BNN.

3. The CPE is defined to be present if the gradient of the test loss wrt $\lambda$ is negative at $\lambda=1$.

Thus, given 2, if criterion 3 is satisfied then so is 1, i.e. the presence of a CPE implies underfitting. My confusion as a reader is that 2 is a given, and therefore a CPE seems synonymous with underfitting. So for a BNN, is a CPE the same as underfitting _by definition_ or does underfitting _cause_ as CPE? I'm not sure I've posed this question clearly, but would appreciate any insight from the authors on the subject.



### References
(Adlam et al., 2020) Adlam, B., Snoek, J. and Smith, S.L., 2020. Cold posteriors and aleatoric uncertainty. arXiv preprint arXiv:2008.00029.

(Zeno et al., 2020) Zeno, C., Golan, I., Pakman, A. and Soudry, D., 2020, November. Why cold posteriors? on the suboptimal generalization of optimal bayes estimates. In Third Symposium on Advances in Approximate Bayesian Inference.

---

> ### Author Response · Authors · 2024-06-05
> **Response for Reviewer GSTQ**
>
> We thank the reviewer for the valuable feedback.
>
> # Weaknesses
> > While the majority...
>
> Thank you for the references. Zeno et al 2020 is really relevant here. We were not aware of it. We have added and discussed them in the revision under relevant paragraphs and adjusted our claims accordingly.
>
> > The paper recognises...
>
> We agree that our proposed method does not directly involve tuning the temperature based on the training data, which deviates from the typical usage of Empirical Bayes (EB). In light of this, we will remove references to Empirical Bayes from the article to ensure clarity and accuracy in our descriptions. We appreciate your input on this matter.
>
> # Minor Weaknesses
>
> > I definitely agree...
>
> We thank the reviewer for noticing, and we fully agree that this definition only serves as an initial step. We hope that it will pave the way for further investigation in future research. As such, we have added this as a limitation of the current work.
>
> > The argument in...
>
> Thank you for bringing up this interesting discussion. We have added them in the corresponding sections. In particular, we added a discussion about the overall theoretical implications of Theorem 4. More precisely:
> 1. As the reviewer already noticed, A can be demonstrated by comparing Figs 3 and 4. We added a discussion about this point.
> 2. Interestingly, B can be demonstrated in Fig 2(c) and 2(d): in both cases, we have a little flexible model (i.e., a linear model) and a very small training set (i.e., 5 samples), which indeed incurs a CPE, as commented by the reviewer.
> 3. This implication can also be demonstrated in the linear regression (i.e., a little flexible model) too using a larger training dataset (i.e, 50 samples). As shown in the new Fig 2(e), in comparison with Fig 2(c), there is virtually no CPE, as noticed by this reviewer.
>
> Again, thanks for bringing this up. Very insightful comment!
>
> # Clarification
>
> > I would be...
>
> It is correct that the presence of a CPE implies underfitting. However, they are not equivalent. If they were equivalent, the statement "If there is no CPE, there is no underfitting" should hold true. However, it's important to note that underfitting can be addressed in various ways, not necessarily by tempering. We also note that Proposition 3 shows that your point 2 is not necessarily given or guaranteed, because it may happen that the empirical Gibbs loss is really at the bottom (something very close to interpolation, a common phenomenon in neural networks). Therefore, the absence of CPE does not guarantee the absence of underfitting. As such, they are not equivalent, and underfitting includes more scenarios than CPE. Additionally, it's essential to clarify that the argument is a logical derivation rather than a causal one. In other words, we do not claim a causal relationship between them. In fact, it is challenging to establish causation as both underfitting and CPE can be influenced by other factors, such as misspecification and relative model size to sample size.
>
> References
> 1. (Adlam et al., 2020) Adlam, B., Snoek, J. and Smith, S.L., 2020. Cold posteriors and aleatoric uncertainty. arXiv preprint arXiv:2008.00029.
> 2. (Zeno et al., 2020) Zeno, C., Golan, I., Pakman, A. and Soudry, D., 2020, November. Why cold posteriors? on the suboptimal generalization of optimal bayes estimates. In Third Symposium on Advances in Approximate Bayesian Inference.
>
> # Critical
> > Include references...
>
> > Either justify why...
>
> Thanks for the comments. We have included the references and removed the Empirical Bayes view in our article.
>
> # Minor
>
> > For completeness...
>
> We understand that the test accuracy results maybe helpful. However, our results are on log-loss but not 0-1 loss and including accuracy results will double the number of figures. We will add the results of the settings if you are interested in the appendix.
>
> > Experimentally validate Theorem 4...
>
> A discussion about the impact of training set size in CPE, validating Thm 4 has been introduced at the end of Section 5.
>
> > Show results...
>
> We note that the purpose of Figures 3 and 4 is to illustrate how one factor influences the Cold Posterior Effect (CPE) when the other factor is fixed. As such, we did not include the scenario where both factors are changed simultaneously, as the effect can be inferred from the cases already presented. However, we are open to adding this case (maybe in the appendix) if the reviewer believes it would enhance clarity in the discussions.
>
> # Very Minor
>
> > power of $\lambda$
>
> We’ve made updates to the new version.
>
> > Section 2: "If it...
>
> We’ve made updates to the new version.
>
> > Section 3: Should eqn...
>
> It should be Proposition 2, we do not have Proposition 1 but Definition 1.
>
> > Section 3: "We will...
>
> We’ve made updates to the new version.
>
> > Square brackets...
>
> We’ve made updates to the new version.
>
> > Section 4 repetition...
>
> We’ve made updates to the new version.

---

> > ### Author Response · Authors · 2024-06-05
> > **Response for Reviewer GSTQ, part 2**
> >
> > # Very Minor
> >
> > > Section 4: "...
> >
> > We’ve made updates to the new version.
> >
> > > Section 6: Do not...
> >
> > We’ve made updates to the new version.
> >
> > > The mathematical results...
> >
> > We provide an intuitive way to see it. If the loss function is more strongly correlated with the expected Gibbs loss, it could be seen to have more data or more information of the data from the (thus “meaningful” data augmentation ) true data-generating process. Hence, the capacity of data increases compared to the capacity of models, resulting in more significant underfitting.

---

> > > ### Comment · Reviewer_GSTQ · 2024-06-06
> > >
> > > Thank you for your response and the updated manuscript. I believe this is an improvement in general.
> > >
> > > I do not feel the Section Overview boxes are necessary or helpful, but will wait for other reviewers to comment and am willing for them to be kept if other reveiwers feel strongly.
> > >
> > > ### Zeno et al., 2020
> > > Thanks for including this citation. I agree with the statement in Section 4, that you are extending their results beyond classification. However, I believe Contribution (ii) should either be removed or amended to state that you have extended existing work to show that tempered posteriors are legitimate posteriors _in the general case_, not just for classification.

---

> > > > ### Comment · Reviewer_GSTQ · 2024-06-12
> > > >
> > > > I should have stated previously: overall I am ready to recommend acceptance.

---

> ### Author Response · Authors · 2024-06-13
>
> Thank you for the prompt reply and positive feedback. We have updated Contribution (ii) to "We show in a more general case that any tempered posterior is a proper Bayesian posterior with an alternative likelihood and prior distribution in Section 4, extending \cite{zeno2021why} in the case of classification."

---

### Review · Reviewer_hcmZ · 2024-05-18

**Summary Of Contributions:**

The paper provides a theoretical and empirical study of the cold posterior effect (CPE). The authors offer both theoretical and practical evidence supporting the main argument of the work: that CPE originates from underfitting. Additionally, the authors review their connection to existing arguments regarding the effects and origins present in the field. For instance, they demonstrate that likelihood and prior misspecifications are relevant to CPE only in cases showing underfitting. Furthermore, the paper argues that data augmentation is a cause of CPE for the same reason—underfitting of the models.

**Audience:**

Yes

**Claims And Evidence:**

Yes

**Requested Changes:**

A Critical suggestion:
Figures 3 and 4 have G(p^{\lambda}), which is not shown anywhere and is clearly a typo/mistake.

Minor Suggestions / Recommendations:

Overall, captions need more descriptions, with more focus on how they relate to discussions in the corresponding sections (especially figures 3, 4, and 5).

Since this paper provides an extensive discussion and comparisons between existing methods and authors' theoretical framework, it would be convenient to have images on the same page as sections discussing them for convenience's sake.

**Strengths And Weaknesses:**

Strengths:
The main strength of the paper stems from the theoretical analysis, which is later supported by experimental evidence. The theoretical analysis is sound, and the experiments align with the research objectives.

The overall writing style is appropriate (even though occasionally relaxed) and the experiments are clearly presented. Additionally, a significant effort was made to connect existing, somewhat disjointed explanations and approaches in CPE research, enhancing the work's value to researchers in the field and knowledge in general.

Weaknesses:

The main weakness of the paper is that, despite the in-depth discussions, it is quite hard to follow the lines of reasoning. I highly suggest emphasizing your hypothesis (as in Wenzel et al., 2020) or the main argument (in bold or italics), providing the discussion (like in sections 4,5 and 6), and then the conclusion within the same section or even within subsections (likelihoods, priors, model misspecification). The use of lists and tables with the results of the discussion is encouraged. Parts of discussions could be trimmed down and put into the Appendix section.

The same goes for images. For instance, figures 3a and 3b are primarily related to the discussion in section 5 on likelihood misspecification, but figure 3c is relevant to the discussion on priors instead. It was not clear how it related to the discussion at first, but it becomes clearer later in the text.

Definition 1, despite being very relevant to the analysis, would benefit from mentioning similar ways of defining CPE or similar quantities in the literature. The way it is defined here, and the authors acknowledge that it’s not “[the authors] do not claim this is the best possible formal characterization,” would benefit from referring to existing cases of definitions of such quantities.

In section 5, “Model size, CPE, and underfitting,” the experiments demonstrate and authors argue that larger CNN models exhibit less CPE. However, Wenzel et al., 2020 in figure 11 showed that larger models tend to possess much stronger CPE compared to either shallower or narrower version of the same architectures. Some discussion on that would be particularly relevant and interesting.

---

> ### Author Response · Authors · 2024-06-05
> **Response for Reviewer hcmZ**
>
> We thank the reviewer for the valuable feedback.
>
> > The main weakness of the paper is that...
>
> We’ve added high-level summaries for each section to make them clearer.
>
> > Images...
>
> Regarding the images, we’ve added more descriptions in the captions to make them clearer.
>
> > Other definitions of CPE
>
> To the best of our knowledge, we are not aware of any formal definitions. The convention of CPE is usually, testing loss decreases wrt to \lambda or temperature $T$.
>
> > In section 5, “Model size, CPE, and underfitting,”...
>
> Thanks a lot for pointing us to this figure. We did not realize this experiment was present there. One of the possible explanations is that they are using full-tempering while we are using likelihood tempering.
> 1. For full-tempering, Theorem 2 does not necessarily hold anymore. Intuitively, since \lambda works on both likelihood (data) and prior (regularization) together, the effect of increasing \lambda is mixed, thus not necessarily improve the fit on training data.
> 2. Thus, the CPE brought by full-tempering, as \lambda increases, does not necessarily comes with a better training loss, i.e., training loss may not be improvable. Hence, full-tempered CPE can no longer be interpreted as just underfitting only.
> 3. As such, for full-tempering, increasing the model capacity may not achieve a lower degree of CPE as in our case.
>
> > A Critical suggestion...
>
> We’ve updated the figures.
>
> > Overall, captions need more descriptions
>
> Captions. We’ve updated the captions to make them clearer.
>
> > Positioning of the figures.
>
> We’ve made efforts to position the figures better.

---

> ### Comment · Reviewer_hcmZ · 2024-06-11
>
> Thanks for updating the manuscript and for your response! I would suggest including a couple of sentences of discussion as you described it ("For full-tempering, Theorem 2 does not necessarily hold anymore....") regarding the model size effects and the difference between your work and the work of Wenzel et al., 2020. Maybe in section 5.5, where I think it is most appropriate.
>
> Overall, I am ready to recommend acceptance.

---

> > ### Author Response · Authors · 2024-06-13
> > **Response for Reviewer hcmZ**
> >
> > Thanks again for the quick reply and the positive feedback! We've added the discussion in section 5.5 as you suggested. Please check the updated version.

---

### Review · Reviewer_WSjK · 2024-05-21

**Summary Of Contributions:**

The paper investigates the cold posterior effect (CPE) and argues that underfitting is a necessary condition for the CPE to occur. The authors study how tempering the likelihood changes a parametric model's training and test losses. They show that more tempering always reduces the training error; hence, if the CPE occurs and tempering reduces the test error too, the model must have underfitted the data. In light of this finding, the authors investigate and reinterpret some proposed causes of CPE, such as data augmentation, model misspecification, and errors with the approximate inference procedure.

**Audience:**

Yes

**Claims And Evidence:**

No

**Requested Changes:**

In order for me to recommend the paper's acceptance, the authors need to significantly improve their writing and address the issues I outlined in the weaknesses section.

Besides this, I have a few more minor requested changes:
 - The notation $[n]$ is undefined; I believe it means the integers from $1$ to $n$.
 - The reconstruction term in eq 3 is missing a $1/n$ term.
 - Throughout the paper, the authors refer to the derivative of many quantities with respect to $\lambda$ as the "gradient." While this is technically correct, I think it is confusing as $\lambda$ is always a scalar. Hence, I would prefer if the authors just called it a derivative instead.
 - The authors use $p^\lambda$ to denote the tempered posterior. I think this needs to be clarified; I recommend using $p_\lambda$ to avoid confusion with raising to a power.
 - Above prop 3: I have not heard the minimum be called the floor before; I think it would make more sense to stick with the standard terminology.
 - Proof of proposition 7: The fourth-to-last line has an extra $\alpha$
 - Proof of prop 7: $\bar{p}$ should be defined in the theorem statement
 - I think it is confusing to call $p^\lambda$ the **Bayes** posterior (e.g. in insight 1), before the authors explain precisely what they mean in section 4.
 - "In other words, when such a Bayesian posterior contains more information about the data-generating distribution, it continues to fit the originally provided training data in a similar manner" - I'm not sure what this sentence means.
 - $d\theta$ is missing in the appendix in many integrals.
 - Prop 5: inconsistent notation: $q$ in the main text, $p'$ in the appendix; proof in an appendix not quite right because $p'$ is not a valid prior since it doesn't integrate to $1$. Also, it does not prove the same statement since $p'$ doesn't depend on the input location in the appendix.
 - "However, the standard likelihoods used in deep learning, like softmax or sigmoid, implicitly assume a higher level of aleatoric uncertainty in the data." - I'm not sure what this statement means.

**Strengths And Weaknesses:**

# Strengths
I must note that I am not intimately familiar with the literature on the CPE, so I cannot evaluate if some of the claimed contributions have appeared elsewhere.

That said, I think the authors' work represents a reasonable step towards better understanding the CPE. The central premise of the authors' work rests on the identity in Proposition 2, which shows that tempering the likelihood always decreases the training error. While this is certainly not surprising, it is valuable to connect it to underfitting.

Similarly, while the observation that the tempered posterior can be written as a posterior arising from performing Bayesian inference using a different prior and likelihood function is fairly trivial, it is valuable to draw attention to this fact. I particularly liked the observation that tempering can make the prior depend on the training input locations.

Furthermore, I found most of the paper to be reasonably well-written.

I have also carefully checked the supplementary material for the proofs and found all claims to be correct under reasonable assumptions (see weaknesses for more on this).

# Weaknesses
Overall, I have three main worries regarding the paper:
 1. The originality of the theoretical claims
 2. The strength of the theoretical analysis
 3. The exposition in Sections 4-6

## Originality

While I found the authors' theoretical claims correct, they all follow from direct and fairly elementary calculation; hence, I wonder if these identities have been derived elsewhere in the machine learning or statistical physics literature before. While I don't know a source that contains the exact identities the authors derive in this paper, all of their results follow mutatis mutandis from the derivations in [1]. I have the same worry regarding the contents of section 4.

I am not suggesting that the paper be rejected if these results are not original. However, I think the authors should at least note that very similar results already exist in the literature (although in a different context, perhaps) and reference these.

## Strength of the theoretical analysis

There are a few points that I think, if clarified, would significantly strengthen the paper:
 - The authors should state under what assumptions their results hold. While it's clear that the results should apply to "nice enough" models, it is not clear what "nice enough" means. For example, it is not even clear under what assumptions the tempered posterior would exist as the tempered posterior might not be normalizable.
 - The theory's limitations are unclear, and the analysis could be sharper. For example:
what are the limitations of Definition 1?
   - Are saddle points in lambda possible?
   - Under what assumptions can we find an optimal lambda?
   - Is the Bayes loss convex in lambda?
   - Is it possible that under some circumstances, the Bayes loss monotonically decreases in lambda?
   - Currently, the authors only look at the behavior of the loss for a fixed set of training data; what happens, e.g., in the limit as we collect more and more data?
 - It is unclear what assumptions about the data-generating process are needed where. For example, I presumed that all arguments went through for non-iid data, but Proposition 5 seems to assume that the data is, in fact, iid. It is particularly important to clarify the context of section 6, which deals with data augmentation. The CPE has been linked to data augmentation, creating correlated training data [2], which the authors do not discuss in Section 6. Could the authors comment on this?

## Sections 4 - 6
I think the writing in these sections could be improved. They feel longer than needed and contain many references that make them hard to follow.

The examples in Section 4 take up a lot of space, and it is unclear what purpose they serve; if the authors meant them as a set-up for the experiments in Sections 5 and 6, then they should merge them appropriately instead.

I found Section 5 especially difficult to follow. This is in part because the paragraphs' focus is unclear: they combine some high-level technical discussion with a literature review and interpret the authors' experimental results. To be blunt, I couldn't really figure out the authors' take-home messages in this section.

Finally, in Section 6, besides my worry about the lack of discussion on the iid assumption on the training data, it also needs to be clarified how Sections 6.1 and 6.2 relate to each other. Much of the content in these sections seems almost duplicated, and I'm wondering what insight they are supposed to provide.

# Questions
 - Figs 3 and 4: where is the green curve for $G(p)$?
 - Figs 5 & 6: why are the augmented curves worse than the non-augmented ones?

# References

 - [1] Masrani, V., Le, T. A., & Wood, F. (2019). The thermodynamic variational objective. Advances in Neural Information Processing Systems, 32.
 - [2] Bachmann, G., Noci, L., & Hofmann, T. (2022). How tempering fixes data augmentation in bayesian neural networks. arXiv preprint arXiv:2205.13900.

---

> ### Author Response · Authors · 2024-06-05
> **Response for Reviewer WSjK**
>
> We thank the reviewer for your valuable feedback.
>
> # Weaknesses
> ## Originality
>
> Thanks for bringing this up. We agree that the derivation of the identities isis simple and could be already derived somewhere else, but, at the same time, our contribution is in the application of these identities in the case of CPE. We checked reference [1] and their derivations are different. They take gradients wrt lambda, but lambda in [1] corresponds to variational and model parameters. Even though, Eq (11) in [1] could probably be generalized to cover the results in [1] and in this paper.
> About Section 4, as noted by reviewer  GSTQ, there was a published paper with similar results. See the answer we give to GSTQ. We have introduced this discussion in the paper.
>
> ## Strength of the theoretical analysis
>
> > The authors should state under what assumptions...
>
> We agree this is a fair criticism. The main underlying assumption is that the tempered posterior is always a proper distribution. We will include this in this work. At the same time, please note that if supervised classification, we always have that $p(D|\theta) \leq 1$. In consequence, for any $\lambda>0$, we have
> $1 = \int \pi(\theta) d \theta > \int p(D|\theta)^\lambda \pi(\theta) d \theta $
> So, the normalization constant of the tempered posterior always exists in this case. We have updated the manuscript with this discussion in Section 2.
>
> > The theory's limitations are unclear, and the analysis could be sharper. For example: what are the limitations of Definition 1?
>
> The characterization of CPE we propose is defined only as the local change of Bayes loss at $\lambda = 1$. This approach does not account for scenarios where significant decreases in Bayes loss at other \lambda values might also indicate the presence of CPE. We will add this to the limitation sections of the paper to make it clear in the main text.
>
> > Are saddle points in lambda possible? Under what assumptions can we find an optimal lambda?
>
> The presence of saddle points would be possible, where the derivative at $\lambda = 1$ is zero but there exists an optimal $\lambda < 1$.
>
> > Is the Bayes loss convex in lambda?
>
> Regarding the convexity of the Bayes loss, it does not hold in general, see Figures 2(b), 5(c), 6(c) for example.
>
> > Is it possible that under some circumstances, the Bayes loss monotonically decreases in lambda?
>
> Regarding its monotonicity, we can look at Figure 2. In this scenario, it is clear that having a monotonically decreasing Bayes loss is not ideal as it means that the model may be underfitting too much.
>
> > Currently, the authors only look at the behavior of the loss for a fixed set of training data; what happens, e.g., in the limit as we collect more and more data?
>
> Thm 4 can be used to reason about what happens as we approach infinite data. If we have many data points we expect $\bar{p}_{\lambda=1}$ = $p_{\lambda=1}$ and, then, we should not see CPE. We have added a discussion on this in Section 5.5 on the main text, also,  as an answer to a question raised by Reviewer GSTQ.
>
> > It is unclear what assumptions...
>
> Thanks for raising this point. We have realized that Proposition 5 does not need an i.i.d. assumption, it only needs that the likelihood can be fully factorized. It was a misunderstanding on our side. We have fixed the proof and made it clear in Section 2 that we are assuming that the likelihood fully factorizes.
>
> We also thank the reviewer for bringing the connection with [2]. We have given new thought to this work and we have realized our work aligns with the findings of [2] about the degree of model invariances for the data augmentation. For example, if the model is completely invariant to the data augmentation, we would have that \hat{L}_{DA}(D,\theta)=\hat{L}(D,\theta). And, in consequence, p_{\lambda=1}^DA=p_{\lambda=1}  and the inequality of Eq. (15) will be an equality. I.e., DA would not cause a stronger CPE. If there is no CPE under the standard Bayesian posterior (i.e., $\lambda=1$ is optimal), using DA over a completely invariant model would not induce any CPE (i.e., $\lambda=1$  would be optimal as well). We have not had the time to elaborate on this, but we are working to include this discussion in the next version of the paper.

---

> > ### Author Response · Authors · 2024-06-05
> > **Response for Reviewer WSjK, part 2**
> >
> > ## Sections 4 - 6
> > Thank you for your feedback. We have made changes and revised these sections to be more concise and clearer.
> > 1. In section 4, we would like to show that tempered posteriors are proper Bayesian posteriors, and thus tempering is well-aligned with the Bayesian paradigm: tempering implicitly finds posteriors with less misspecified likelihoods and priors, and as a result, is less underfitted and thus brings better performance.
> > 2. In section 5, we agree with your comments and we’ve restructured the content for a better presentation.
> > 3. In section 6, the iid assumption is not needed for our analysis (please refer to the previous answer). In section 6.1, instead of the Bayes loss, we first present the analysis of Gibbs loss because it’s easier to interpret and understand. Also, it serves as the upper bound for the Bayes loss.
> >
> > # Questions
> >
> > > Figs 3 and 4...
> >
> > For Fig. 3 and 4, we mistakenly added the wrong legend $G(p)$, we’ve removed them in the new figures.
> >
> > > Figs 5 & 6...
> >
> > For Fig. 5 and 6, we would like to emphasize that the CPE is more significant with data augmentation (the change of the Bayes loss as \lambda increases). As for the performance, we believe it depends on a lot of factors, i.e., the way of tempering, the way of augmenting data, models, and data.
> >
> > # Requested Changes
> >
> > > The notation...
> >
> > We’ve made updates to the new version.
> >
> > > The reconstruction term...
> >
> > Please note that the $1/n$ term is also not in the KL term, thus making the reconstruction term (over all the data) valid.
> >
> > > Throughout the paper...
> >
> > We’ve made updates to the new version.
> >
> > > The authors use $p^\lambda$...
> >
> > We’ve made updates to the new version.
> >
> > > Above prop 3...
> >
> > We’ve made updates to the new version.
> >
> > > Proof of proposition 7...
> >
> > We’ve made updates to the new version.
> >
> > > Proof of prop 7...
> >
> > We’ve made updates to the new version.
> >
> > > I think it is confusing...
> >
> > Actually, $p_\lambda=1$ is the Bayesian posterior. We’ve in introduced this in sec. 2.2, the paragraph above eq.3.
> >
> > > "In other words..."
> >
> > We’ve made updates to the new version.
> >
> > > $d\theta$ is missing...
> >
> > We’ve made updates to the new version.
> >
> > > Prop 5: inconsistent notation...
> >
> > We’ve made updates to the new version.
> >
> > > "However, the standard likelihoods..."
> >
> > We’ve made updates to the new version.

---

> > > ### Comment · Reviewer_WSjK · 2024-06-07
> > > **Response to the authors**
> > >
> > > I thank the authors for their rebuttal. Overall, I am happy with the improved manuscript, and I am ready to recommend acceptance. I quite like the section overview boxes they added.
> > >
> > > That said, I think the quality and value of the paper could be significantly improved by formulating the theoretical statements more carefully and rigorously. While the authors have made some clarifications in the updated manuscript, there are still large gaping holes in what the necessary assumptions are for the results to hold. I don't think this should inhibit the paper's acceptance, as the results should hold when all involved quantities are "nice enough."
> > >
> > > Among others, the most pressing issues I found are:
> > >  - The authors state no assumptions on the sample space $\mathcal{Y} \times \mathcal{X}$, but in the rest of the paper, they clearly assume that at least $\mathcal{Y}$ is either a discrete set or a subset of $\mathbb{R}^n$. While I don't contest that these cases have the greatest practical relevance, the authors should note this restriction explicitly or clearly state the necessary assumptions for their results to hold. This imprecision later leads to technically meaningless results as well, for example, in Proposition 6, which states that $\frac{d}{d\lambda}H[q(y \mid x, \theta, \lambda)] \leq 0$. However, from its statement, it's not clear whether the authors mean Shannon entropy or differential entropy. Indeed, in the appendix, they assume it is the differential entropy, meaning that it does not imply their conclusions for the classification example.
> > >  - "We are then assuming that the likelihood function fully factorizes, $p(D \mid \theta) = \prod_{x, y} p(y \mid x, \theta)$." - The authors write such an equality in multiple places in the updated manuscript. I think I see what the authors mean here, but this equality cannot be true in general, as $p(D \mid \theta)$ is a distribution over $(\mathcal{Y} \times \mathcal{X})^n$, while $\prod_{x, y} p(y \mid x, \theta)$ is a distribution over $\mathcal{Y}^n$ only. If the authors say this because they want to focus on the distribution of $y$s, then they should state this explicitly and change the equality to proportionality. Otherwise, they should clarify what they mean to avoid confusion.
> > > - "However, the standard likelihoods used in deep learning for image classification, like softmax or sigmoid, implicitly assume a higher level of aleatoric uncertainty in the curated image data..." - Again, it is unclear what the authors mean by this exactly. The authors should formally specify what they mean with the implicit bias of softmax/sigmoid towards high aleatoric uncertainty.

---

> > > > ### Author Response · Authors · 2024-06-14
> > > >
> > > > Thank you for the detailed feedback and suggestions!
> > > >
> > > > We have updated the article accordingly.
> > > >
> > > > - In section 2.1, we specify the assumptions on the spaces, assumptions about the factorized likelihood, and clarify the notation of $p(D\mid \theta)$. In particular, we consider a finite set Y in supervised classification problems and consider Y as a subset of $\mathbb{R}$ when facing regression problems. For simplicity, we will also assume that X is a subset of $\mathbb{R}^d$.
> > > > - For proposition 5 and 6, we also clarify the assumptions for the elements used in the statements. Proposition 6 verifies under supervised classification with Shannon Entropy, and verifies in regression settings with differential entropy, assuming that Leibniz integral rule holds. Thank you for pointing out the proof is made for differential entropy. The case of Shannon entropy also follows naturally by exchanging the integrals by summations, where no integral rule is needed due to the linearity of the derivative operator. We add such a discussion to the proof as well.
> > > > - We have changed the sentence to the following “However, as previously discussed in \citep{Ait21,KMIW22}, the standard likelihoods used in deep learning for image classification, like softmax or sigmoid, tend to allocate more spread out probabilities to the outcomes, implicitly reflecting a higher level of aleatoric uncertainty.” Together with the previous and the later sentences, the relative meaning of “higher” and “lower” aleatoric uncertainty should hopefully be more clear. In the next paragraph, we also give more concrete examples to make such an idea more precise.

---

> > > > > ### Comment · Reviewer_WSjK · 2024-06-14
> > > > > **Response to the authors**
> > > > >
> > > > > I thank the authors for their reply. I am happy with the changes and believe that the paper would be a good contribution to the literature.

---

### Decision · Action_Editor_1qas · 2024-06-22

**Recommendation:** Accept as is

**Comment:**

The reviewers agreed that the paper should be accepted. However, it has also been suggested that the description of the theoretical assumptions of the work could be improved, so I would recommend the authors take that into account for the camera-ready version.

**Audience:**

The paper should be interesting to some part of the TMLR audience, especially researchers studying the cold posterior effect.

**Claims And Evidence:**

The reviewers agree that this work provides and interesting and coherent perspective on the cold posterior effect and that the claims are well supported.